# Understanding Softmax Attention Layers: Exact Mean-Field Analysis on a Toy Problem

**Elvis Dohmatob**[1,2,3]
[1]Concordia University    [2]Mila–Quebec AI Institute    [3]Meta
elvis.dohmatob@concordia.ca

## Abstract

Self-attention has emerged as a fundamental component driving the success of modern transformer architectures which power large language models (ChatGPT, Llama, etc.) and various other types of systems. However, a theoretical understanding of how such models actually work is still under active development. The recent work of (Marion et al., 2025) introduced the so-called "single-location regression" problem, which can provably be solved by a simplified self-attention layer but not by linear models, thereby demonstrating a striking functional separation. A rigorous analysis of self-attention with softmax for this problem is challenging due to the coupled nature of the model. In the present work, we use ideas from the classical random energy model in statistical physics to analyze softmax self-attention on the single-location problem. Our analysis yields exact analytic expressions for the population risk in terms of the overlaps between the learned model parameters and those of an oracle. Moreover, we derive a detailed description of the gradient descent dynamics for these overlaps and prove that, under broad conditions, the dynamics converge to the unique oracle attractor. Our work not only advances the understanding of self-attention but also provides key theoretical ideas that are likely to find use in further analyses of even more complex transformer architectures.

## 1 Introduction

Understanding the theoretical foundations of transformer layers [Bahdanau et al., 2015] (also see [Schmidhuber, 1992]), particularly self-attention (SA) [Vaswani et al., 2017], remains a critical and largely unresolved challenge in machine learning. SA stands as a cornerstone of modern large language models (ChatGPT, Llama, Gemini, Mistral, Deepseek, Cluade, etc.), driving their unprecedented success across diverse tasks. Unlike classical layers such as feedforward, convolutional, or recurrent architectures, SA enables capabilities that these traditional mechanisms cannot replicate, including efficient capture of long-range dependencies and context-aware representations in a single forward pass. While classical layers benefit from decades of rigorous analysis and well-established theory, the inner workings of SA remain poorly understood, with only a sparse body of research attempting to unravel its operational principles. Moreover, the training dynamics of SA, or how it evolves during optimization, present additional complexities that are crucial for developing more robust, efficient, and interpretable models. A deeper theoretical understanding of SA and its learning behavior is therefore essential not only to explain the empirical successes of transformers but also to unlock their full potential and inform future advancements in neural network design.

Recently, Marion et al. [2025] have considered a simplified version of SA and showed that unlike linear models, it can solve a so-called *single-location regression problem:* the *teacher* model $f_*$ sees an incoming input $X = (x_1, \ldots, x_L)$ made of the embedding vectors $x_\ell$ in $\mathbb{R}^d$ for a sequence of $L$ tokens (e.g words), and must correctly locate a secret block $x_{\ell_*}$ at a random secret index $\ell_* \in [L]$. Refer to Figure 1. This token index is special in that except for additive Gaussian noise,

39th Conference on Neural Information Processing Systems (NeurIPS 2025).

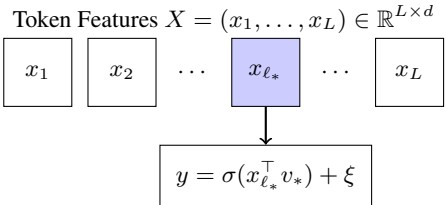

Token Features $X = (x_1, \ldots, x_L) \in \mathbb{R}^{L \times d}$

$$y = \sigma(x_{\ell_*}^\top v_*) + \xi$$

Figure 1: **The Single-Location Regression Problem.** Each $x_\ell \in \mathbb{R}^d$ corresponds a token embedding. The embedding for the secret token at index $\ell_*$ contains signal aligned with a hidden vector $u_*$. The embeddings for tokens at all other indices are pure Gaussian noise. The label $y$ is computed using only this secret token, and all other tokens are ignored. Optionally, we also introduce a link function $\sigma$ to capture non-linear problems ($\sigma$ was taken to be the identity function in [Marion et al., 2025]).

it is aligned with an unknown unit-vector $u_*$ which can be thought of as encoding the position of $\ell_*$; all the other blocks $x_{\ell \neq \ell_*}$ are Gaussian noise. Once this block is identified, the model must then approximate the output given by $y = f_*(x) := x_{\ell_*}^\top v_*$, where $v_*$ is another unit-vector perpendicular to $u_*$. Thus, presumably, the model must somehow figure out the directions $u_*$ and $v_*$ in order to solve this problem. This problem captures some aspects of the sparse parity problem, except it is considerably simpler; for example, it does suffer from the the well-known exponential query complexity lower-bound which characterizes the latter problem. Marion et al. [2025] considered a simplified transformer model

$$\textbf{Pointwise Self-Attention: } g(X; u, v) = \sum_{\ell=1}^{L} p_\ell x_\ell^\top v, \text{ with } p_\ell = \theta(\lambda x_\ell^\top u), \tag{1}$$

for some point-wise activation function $\theta$ and inverse temperature parameter $\lambda$. The parameters of this *student* model are a pair of unit-vectors $(u, v)$.

Going beyond [Marion et al., 2025] which considered pointwise/separable SA (1), we consider softmax SA which better reflects what is actually used in transformers. Our student model is then

$$\textbf{SoftMax Self-Attention: } f(X; u, v) = \sum_{\ell=1}^{L} p_\ell \sigma(x_\ell^\top v), \text{ with } p_\ell = \frac{e^{\lambda x_\ell^\top u}}{\sum_{\ell'=1}^{L} e^{\lambda x_{\ell'}^\top u}}. \tag{2}$$

In our extension, we also include a possibly nonlinear link function $\sigma$ (known to the learner) used to compute the labels using the the embedding of of a token at a secret index $\sigma(x_{\ell_*}^\top v_*)$ (instead of $x_{\ell_*}^\top v_*$) for the true labels, and $\sigma(x_\ell^\top v)$ (instead of $x_\ell^\top v$ as in (1)) for the values in our attention model (2). This link function should not be confused with the softmax layer which is always present in the setting we consider in our work. These are two separate extensions of (1).

Importantly, our theoretical analysis is valid for all $L$ up to a limit which is super-polynomial in the dimension $d$, i.e., $\log L = O(d)$. In contrast, the analysis of Marion et al. [2025] is only limited to sub-linear number of blocks $L = o(d)$.

**Main Contributions.** Our contributions can be summarized as follows:

– *Exact Analytic Formulae for the Risk.* In an appropriate asymptotic scaling regime for $d$ and $L$ (refer to (9)), we obtain precise analytic expressions for the population risk of our softmax self-attention model (2) (Proposition 1 and Proposition 2). Our approach uses ideas from the classical analysis of the Random Energy Model (REM) [Derrida, 1981, Lucibello and Mézard, 2024] to handle the softmax, which maps to the Gibbs distribution induced by the disorder in corresponding REM. In order to incorporate the nonlinearity $\sigma$, we extend a recent result of [Zavatone-Veth and Pehlevan, 2025]. See Proposition 12, Proposition 13, and their corollaries (Appendix C).

– *Optimization Dynamics.* We study the optimization landscape of projected gradient-descent on the population risk relative a manifold corresponding to spherical constraints on the model parameters. We classify the stationary points and show that for a large variety of link functions, the induced dynamics always has the optimal model parameters as an attractor (Propositions 8, 10, and 9).

– *No Need for Special Initialization.* In Proposition 4 we focus on the linear link function $\sigma(t) \equiv t$ and remove a critical initialization assumption which was made in [Marion et al., 2025]. Indeed some

of the main results about optimization in the aforementioned paper assumed that initialization be selected from a peculiar manifold, which effectively assumes some knowledge of the teacher / oracle parameters $(u_*, v_*)$, which is not feasible a feasible requirement in practice.

## 2  Related Work

The self-attention mechanism, introduced by Vaswani et al. [2017] drives much of modern deep learning, notably in natural language processing and computer vision. By adeptly capturing long-range dependencies, it eclipsed recurrent neural networks in many tasks. Empirical breakthroughs like BERT [Devlin et al., 2019] and Vision Transformers [Dosovitskiy et al., 2021] expanded its reach, sparking theoretical exploration of its mechanics.

A sparse literature of studies have unpacked aspects self-attention's expressive power, complexity, and optimization. Schlag et al. [2021] crafted a minimal attention model without sans positional encoding or normalization, to study its core behavior, while Cui et al. [2024] explored a solvable dot-product attention model, identifying a phase transition between positional and semantic learning driven by positional encoding. Another common simplification of attention, is the so-called linear attention mechanism, where the softmax layer is removed altogether. Such models have been intensively studied in the setting of in-context learning to derive theoretical insights on the internal workings of transformers [Ahn et al., 2023, Von Oswald et al., 2023, Zhang et al., 2024, Lu et al., 2024].

The work which is most related to ours is [Marion et al., 2025] which considers a simplified attention model (1) with the softmax layer replaced by a pointwise function, and study the generalization profile and the optimization landscape induced by such models. In contrast, we consider the more difficult (and practically relevant) case of softmax attention and provide a complete theoretical picture.

Let us note that Dong et al. [2021] showed that except if MLP layers and skip-connections are also used, single-layer SA transformers have a strong inductive bias to converge to rank-1 matrices, a simplicity bias which would limit the applicability to complex problems. Fortunately, the single-location problem studied here and in the reference work [Marion et al., 2025] is just simple enough to be captured by a single-layer SA transformer without need of MLP layers or skip-connections.

## 3  Preliminaries

### 3.1  Problem Setup

**Data Distribution.**    Let $d$ and $L$ be positive integers, $\epsilon > 0$ and $\gamma \in (0, 1)$ be real numbers, and consider the following data distribution: $P$ on $[L] \times \mathbb{R}^{L \times d} \times \mathbb{R}$ given by $(\ell_*, X, y) \sim P$ iff

$$\textbf{(Secret Token Index) } \ell_* \sim Unif([L]), \tag{3}$$

$$\textbf{(Secret Token Features) } x_{\ell_*} \sim \mathcal{N}(c\sqrt{d}u_*, \gamma^2 I_d), \text{ with } c := \sqrt{1 - \gamma^2}, \tag{4}$$

$$\textbf{(Dummy Features) } (x_\ell)_{\ell \in [L], \, \ell \neq \ell_*} \overset{iid}{\sim} \mathcal{N}(0, I_d), \tag{5}$$

$$\textbf{(All Features) } X = (x_1, \ldots, x_L) \in \mathbb{R}^{L \times d}, \tag{6}$$

$$\textbf{(Label Noise) } \xi \sim \mathcal{N}(0, \epsilon^2), \text{ independent of } X, \tag{7}$$

$$\textbf{(Label) } y = \sigma(x_{\ell_*}^\top v_*) + \xi. \tag{8}$$

We will denote the marginal distribution of the features $X$ by $P_X$. Here, $u_*, v_* \in S_{d-1}$, where $S_{d-1}$ is the $(d-1)$-dimensional unit-sphere in $\mathbb{R}^d$, while $\sigma : \mathbb{R} \to \mathbb{R}$ is a link function. The unit-vectors $u_*$ and $v_*$ are fixed but unknown to the learner. The constant $\gamma \in (0, 1)$ controls the signal-to-noise ratio (SNR) of the problem. As argued in [Marion et al., 2025] in the limit $\gamma \to 0^+$ the dummy features vanish, and the problem reduces to the usual Gaussian linear signal + noise model, which is solvable via simple linear regression with a large enough sample from $P$. The situation is graphically illustrated in Figure 1.

**The Single-Locator Regression Problem and Softmax Self-Attention.**    As already mentioned in the introduction, the task is to approximate the true label function $X \mapsto \sigma(x_{\ell_*}^\top v_*)$. Of course, neither the sample-dependent index $\ell_*$, nor the unit-vectors $u_*$ and $v_*$ are known to the learner. For a pair

of unit-vectors $(u, v) \in S_{d-1}^2$, consider the following simplified softmax self-attention (SA) model $f$ introduced in (2). The inverse-temperature $\lambda > 0$ controls the sharpness of $f(X; u, v)$. We will impose the following inverse-temperature scaling $\lambda = \beta\sqrt{d}$, with fixed $\beta > 0$. Within this class of models, the one with parameters $(u_*, v_*)$ will be referred to as the *teacher / oracle* model and denoted $f_*$. We shall see in Proposition 2 that this oracle model can indeed approximate the true label function if the feature noise parameter $\gamma$ is not too large. On the other hand, if the learnable parameter vector $u$ is close to the oracle version $u_*$, then the softmax will concentrate its mass around the right index $\ell_*$, allowing the model to select the value $\sigma(x_{\ell_*}^\top v)$ from all the other values $\sigma(x_\ell^\top v)$. Then, if $v$ is itself close to the oracle version $v_*$, the output of the model $f$ will approximate the true labels $\sigma(x_{\ell_*}^\top v_*)$. Thus, goal is to learn the oracle parameter $(u_*, v_*)$.

We work in the following asymptotic regime where $d$ and $L$ are large but $L$ is exponential in $d$, i.e

$$d, L \to \infty, \quad \log(L)/d \to \alpha \in (0, \infty). \tag{9}$$

In our theory, taking the limit $\alpha \to 0^+$ will correspond to the extreme case where $L$ is at most sub-exponential in $d$ (e.g., polynomial, or even constant as in [Marion et al., 2025]).

**Risk / Test-Error.** We will be interested in the average $L_2$-squared error of the parametrized model $f$ defined in (2), relative to the data distribution $P$, i.e

$$R(u, v) := \mathbb{E}_{(X,y)\sim P}[(f(X; u, v) - y)^2] - \epsilon^2, \tag{10}$$

which measure how well the model solves the single-location task. The offset $\epsilon^2$ corresponds to the irreducible error of the Bayes model $f_{Bayes} : X \mapsto \mathbb{E}[y \mid X]$, due to the label noise.

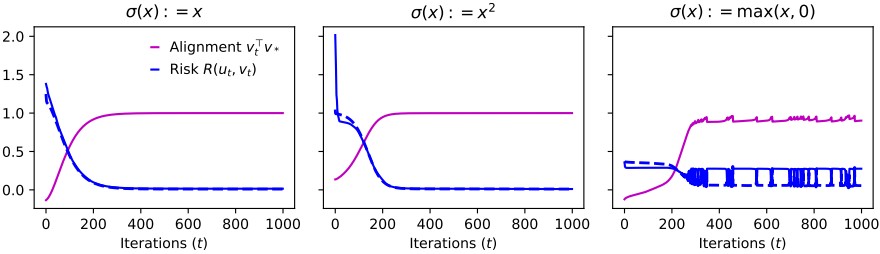

Figure 2: Illustrating the optimization dynamics for various for different choices of link function $\sigma$. For this experiment, we use input-dimension $d = 100$, $L = 20$ blocks, (normalized) inverse-temperature $\beta = 1$, $\gamma = 1/\sqrt{2}$, and label-noise level $\epsilon = 0.1$. The Riemannian gradient-descent scheme is used (29) with step-size $s = 0.01$. The population risk $R$ is replaced by an empirical version $\hat{R} = n^{-1} \sum_{i=1}^n (f(X_i; u, v) - y_i)^2$, where $(X_1, y_1), \dots, (X_n, y_n)$ is an iid sample of size $n = 1000$ from the data distribution $P$. The final risk $R(u_k, v_k)$ shown is evaluated on an independent test sample of size 10000. Broken lines correspond to our theoretical predictions (Proposition 1). Notice the perfect agreement between experiment and our theory. The oscillations in the curves for the ReLU (3rd sub-plot) are reminiscent of the non-smoothness this link function.

## 4 A Mean-Field Approximation

### 4.1 Main Idea: Equivalence to the Random Energy Model

Fix the parameters $(u, v) \in S_{d-1}^2$ of a softmax self-attention model $f$ as defined in (2). For a random data point $(\ell_*, X, y) \sim P$, one can express the output of $f$ as a convex combination like so

$$f(X; u, v) := \sum_{\ell=1}^L p_\ell \sigma(x_\ell^\top v) = p_{\ell_*} f_1(X; u, v) + (1 - p_{\ell_*}) f_2(X; u, v), \text{ with} \tag{11}$$

$$f_1(X; u, v) := \sigma(x_{\ell_*}^\top v), \quad f_2(X; u, v) := \sum_{\ell \neq \ell_*} \frac{e^{\beta\sqrt{d}x_\ell^\top u}}{Z_{-\ell_*}} \sigma(x_\ell^\top v), \tag{12}$$

$$p_\ell := \frac{e^{\beta\sqrt{d}x_\ell^\top u}}{Z} \, \forall \ell \in [L], \quad Z_{-\ell_*} := \sum_{\ell \neq \ell_*} e^{\beta\sqrt{d}x_\ell^\top u}, \quad Z = e^{\beta\sqrt{d}x_{\ell_*}^\top u} + Z_{-\ell_*}. \tag{13}$$

We have isolated the contribution of the secret token index $\ell_* \in [L]$. The other terms (captured in $f_2$) is linked to a $d$-dimensional random energy model (REM) [Derrida, 1981] with $L - 1 = e^{\alpha d}$ configurations with random energy levels $E_\ell = \sqrt{d} x_\ell^\top u$ drawn iid from $\mathcal{N}(0, d)$. It turns out that the mean-field description for such a system is completely captured as follows:

$$\frac{\log Z_{-\ell_*}}{\beta d} \to \psi := \frac{\phi}{\beta} \tag{14}$$

$$\frac{1}{d} \frac{\partial \log Z_{-\ell_*}}{\partial \beta} \to \frac{\partial \phi}{\partial \beta} = r \tag{15}$$

$$\text{where } \phi := \phi(\alpha, \beta) = \alpha + \beta r - s(r) \tag{16}$$

$$r = r(\alpha, \beta) := \min(\beta, \beta_{crit}) = \begin{cases} \beta, & \text{if } \beta < \beta_{crit}, \\ \beta_{crit}, & \text{if } \beta \geq \beta_{crit}, \end{cases} \tag{17}$$

$$\beta_{crit} := \sqrt{2\alpha}, \quad s(r) := r^2/2 \,. \tag{18}$$

**Condensation.** The value $T_{crit} = 1/\beta_{crit} \in (0, \infty)$ is the so-called *condensation temperature*. For temperatures $T = 1/\beta$ less than this value (i.e for $\beta > \beta_{crit}$), the system freezes; only a handful of equilibrium configurations carry the maximum energy, which is of order $\underline{E} := \max_{\ell \neq \ell_*} E_\ell \simeq \psi\sqrt{d}$. Finally, in the limit $\alpha \to 0^+$ in which $L$ is now at most sub-exponentially large in $d$, the parameters $\beta_{crit}$ and $r$ vanish and the the system becomes permanently frozen / condensed, for all values of $\beta$. Our main results for optimization (Section 5) will focus on this regime.

**Remark 1.** *The above is mean-field description in the sense that it ignores corrections of order $1/d$ which could cause statistical fluctuations. Such corrections would require more advanced treatment of the REM, as done in [Bovier and Kurkova, 2004], for example.*

### 4.2 Mean-Field Representation of Models and Their Risk

For theoretical analysis, the softmax in the parametrized model class (2) is troublesome because all the blocks are interacting (via the partition function $Z$). This is precisely the reason Marion et al. [2025] decided to forgo it pointwise / de-correlated self-attention (1) instead.

**Overlaps.** It turns out that in the high-dimensional limit (9), (2) admits a simple description in terms of the "thermodynamic" quantities $r, \phi, \psi$ introduced earlier in (17) and (14), and the overlaps $\mu, \nu, \zeta, \eta, \rho \in [-1, 1]$ defined by

$$\mu := u^\top u_*, \quad \nu := v^\top v_*, \quad \eta := u^\top v_*, \quad \zeta := v^\top u_*, \quad \rho := u^\top v. \tag{19}$$

In particular, $\mu$ and $\nu$ capture the alignment between self-attention transformer $f$ given in (2) with parameters $(u, v)$ and the oracle parameters $(u_*, v_*)$.

Now, the $\mu$ overlap parameter will enter the picture via the following function $p : [-1, 1] \to [0, 1]$ which will play a crucial role in our analysis

$$p(\mu) := \frac{1}{1 + e^{-(c\mu - \psi)\beta d}} \,. \tag{20}$$

This is effectively the high-dimensional limit of the probability that $f$ correctly locates the secret index $\ell_* \in L$ is a random data point $X = (x_1, \ldots, x_L) \sim P_X$.

**Remark 2.** *Notice a sharp phase-transition: for $\mu < \psi/c$, the probability $p(\mu)$ is exponentially close to zero (because the input dimension $d$ is large), while for $\mu > \psi/c$, it is exponentially close to 1.*

**Auxiliary Functions.** Given any $\rho, \zeta \in [-1, 1]$, define an auxiliary function $\sigma_{\gamma,\zeta,\rho} : \mathbb{R} \to \mathbb{R}$ by

$$\sigma_{\gamma,\zeta,\rho}(t) := \sigma(c\zeta\sqrt{d} + \gamma t) - \sigma(\rho r \sqrt{d}) \,. \tag{21}$$

In particular, we write $\sigma_\gamma(t) := \sigma_{\gamma,0,0}(t) = \sigma(\gamma t)$. The so-called "dual kernel" associated with $\sigma_\gamma$, denoted $\bar{\sigma}_\gamma : [-1, 1] \to \mathbb{R}$, will play a crucial role in our results, and is defined by

$$\bar{\sigma}_\gamma(\nu) := \mathbb{E}[\sigma_\gamma(G_1)\sigma_\gamma(G_2)] \,, \tag{22}$$

where $(G_1, G_2) \sim \mathcal{N}_\nu$, a bi-variate Gaussian with unit variance and correlation coefficient $\nu$. For example, for the linear link function $\sigma(t) := t$, we get $\sigma_{\gamma,\zeta,\rho}(t) \equiv \gamma t + (c\zeta - \rho r)\sqrt{d}$ and $\bar{\sigma}_\gamma(\nu) \equiv \gamma^2\nu$.

We shall also need the auxiliary functions $H_1, H_2 : [-1, 1]^3 \to \mathbb{R}$ defined by

$$H_1(\nu, \zeta, \rho) := \mathbb{E}[\sigma_{\gamma,\zeta,\rho}(G_1)\sigma_{\gamma,\zeta,\rho}(G_2)], \quad H_2(\nu, \zeta, \rho) := \mathbb{E}[\sigma_{\gamma,\zeta,\rho}(G_1)\sigma_{\gamma,\rho,0}(G_2)], \quad (23)$$

for $(G_1, G_2) \sim \mathcal{N}_\nu$. Note the implicit dependence of the $H_k$ functions on feature noise level $\gamma$ and the thermodynamic parameter $r$ introduced in (17).

Also define simplified versions $h_1, h_2 : [-1, 1]^2 \to \mathbb{R}$ of the $H_k$'s corresponding to setting $\zeta = 0$, i.e

$$h_k(\nu, \zeta) := H_k(\nu, \zeta, 0). \quad (24)$$

**Remark 3.** *Note that $H_2(\nu, 0, \rho) \equiv H_1(\nu, 0, \rho)$ and $H_k(\nu, 0) \equiv \bar{\sigma}_\gamma(\nu)$ for $k = 1, 2$. Also note that if $r = 0$, then $H_k(\nu, \zeta, \rho) \equiv h_k(\nu, \zeta)$, and the $H_k(\nu, \zeta, \rho)$ doesn't vary with $\rho$.*

Our mean-field analysis will need the following technical condition on the link function $\sigma$.

**Condition 1.** *(A) The link function $\sigma$ is square integrable w.r.t $\mathcal{N}(0, 1)$, and (B) $\sigma$ is positive-homogeneous, meaning that there exists $m > 0$ such that $\sigma(ut) = u^m\sigma(t)$ for all $u > 0$, $t \in \mathbb{R}$.*

*Examples include: linear link function $\sigma(t) := t$; sign link function $\sigma(t) := \text{sign}(t)$; ReLU $\sigma(t) := \max(t, 0)$; quadratic link function $\sigma(t) := t^2$; power link function $\sigma(t) := t^m$ (with $m > 0$); etc.*

The following is one of our main results.

**Proposition 1.** *Suppose Condition 1 prevails. Then, for any model parameters $(u, v) \in S_{d-1}^2$ and for a random data point $(\ell_*, X) \sim P$, it holds in the limit (9) that*

$$f(X; u, v) \simeq p\sigma(x_{\ell_*}^\top v) + (1-p)\sigma(\rho r\sqrt{d}), \quad (25)$$

*where $p = p(\mu)$ is as defined in (20), and $\mu$, $\nu$, $\eta$, $\zeta$, and $\rho$ are as defined in (19).*

*Furthermore, the population risk of the model is given by $R(u, v) \simeq \bar{R}(\mu, \nu, \zeta, \rho)$, where*

$$\bar{R}(\mu, \nu, \zeta, \rho) := p^2 H_1(1, \zeta, \rho) - 2pH_2(\nu, \zeta, \rho) + H_1(1, 0, \rho). \quad (26)$$

*Proof Sketch.* The backbone of the proof (provided in the Corollary 3 of the Appendix) uses ideas from the classical analysis of the REM [Derrida, 1981, Lucibello and Mézard, 2024] to establish (25). In particular, we extend a recent result of Zavatone-Veth and Pehlevan [2025] for our purposes.

Once formula (25) is established, the formula for the risk is a matter of Gaussian integration. $\square$

In order to apply Proposition 1, one needs to compute the auxiliary functions $H_1$ and $H_2$ defined in (23). This is done in Proposition 11 of Appendix B. The said formulae can then be readily exploited to get explicit expressions for the surrogate risk $\bar{R}$ appearing in Proposition 1.

**Linear Link Function.** Consider the special case of the identity link function $\sigma(t) := t$. In this case, thanks again to Proposition 11, we know that $H_k$ functions appearing in Proposition 1 are:

$$\bar{\sigma}_\gamma(\nu) = \gamma^2\nu, \quad H_1(\nu, \zeta, \rho) = \bar{\sigma}_\gamma(\nu) + (a-b)^2, \quad H_2(\nu, \zeta, \nu) = \bar{\sigma}_\gamma(\nu) - (a-b)b, \quad (27)$$

with $a := c\zeta\sqrt{d}$ and $b := \rho r\sqrt{d}$. We obtain the following corollary.

**Corollary 1.** *Under the conditions of Proposition 1, and in the limit (9), it holds that $f(X; u, v) \simeq px_{\ell_*}^\top v + (1-p)\rho r\sqrt{d}$. Moreover, the population risk is given by $R(u, v) \simeq \bar{R}(\mu, \nu, \zeta, \rho)$, where $\bar{R}(\mu, \nu, \zeta, \rho) = (pc\zeta + (1-p)\rho r)^2 d + \gamma^2(p^2 - 2p\nu + 1)$.*

**A Geometric Insight from Corollary 1.** For any fixed $u \in S_{d-1}$, the restriction of the population risk $v \mapsto R(u, v)$ on the set $\{v \in S_{d-1} \mid u^\top v = 0\}$ behaves like a quadratic well $R \simeq (pv^\top u_*)^2 d + \gamma^2\|pv - v_*\|^2$, with deepest point $v(u)$ given by

$$v(u) = (\gamma^2/p)(\gamma^2 I_d + pdu_*u_*^\top)^{-1}v_* = v_*/p \propto v_*.$$

In the above calculation, we have used the Sherman-Morrison formula and the fact that $u_*^\top v_* = 0$. Further, if $cu^\top u_* \geq (1 + \Omega(1))\psi$ (i.e $u$ is within a spherical cap around $u_*$), then

$$p = 1/(1 + e^{-(cu^\top u_* - \psi)\beta d}) = 1/(1 + e^{-\Omega(d)}) \simeq 1 \text{ for large } d,$$

and so $v(u) = v_*/p \simeq v_*$. It is then easy to see that any such $(u, v(u))$ minimizes the population risk $R$. Thus, we only need to get an $\Omega(1)$ alignment of the $u$ parameter with the oracle counterpart $u_*$.

### 4.3 Bayes-optimality of the Oracle Model: A Sharp Phase-Transition

To be sure we are actually in business, we must ensure that the best possible choice of the parameters $(u, v) \in S_{d-1}$ for the parametrized family (2), namely the oracle parameters $(u_*, v_*)$, does indeed achieve zero risk $R = 0$. As the next result shows, it turns out that Condition 2 is a necessary and sufficient condition for this purpose.

**Proposition 2.** *The oracle parameter $(u_*, v_*)$ are indeed optimal for the risk functional $R$ restricted to the parametrized family (2), i.e $R(u_*, v_*) = \inf_{(u,v) \in S_{d-1}^2} R(u, v)$.*

*Moreover, we have the following sharp phase-transition (recall that $c := \sqrt{1 - \gamma^2}$).*

- *(A) If $c \geq (1 + \delta)\psi$ for some $\delta \in (0, 1)$, then in the limit (9), it holds that $R(u_*, v_*) \to 0$. That is, the oracle model with parameters $(u_*, v_*)$ is Bayes-optimal.*

- *(B) If $c \leq (1 - \delta)\psi$, then in the limit (9), it holds that $R(u_*, v_*) = \Omega(1)$, more precisely, $R(u_*, v_*) \to \bar{\sigma}_\gamma(1) > 0$. That is, learning is not possible!*

*Proof.* Indeed, by definition $(u_*, v_*)$ is optimal for the population risk functional $R$. Moreover, thanks to Proposition 1, we have $R(u_*, v_*) \simeq \bar{R}(1, 1, 0, 0)$, with

$$\bar{R}(1, 1, 0, 0) = p^2 H_1(1, 0, 0) - 2p H_2(1, 0, 0) + H_1(1, 0, 0) = (p^2 - 2p + 1)\bar{\sigma}_\gamma(1) = (p - 1)^2 \bar{\sigma}_\gamma(1),$$

where $p = p(1) := 1/(1 + e^{-(c-\psi)\beta d})$. Finally, since $\bar{\sigma}_\gamma(1) > 0$, observe that RHS in the above display vanishes for large $d$ iff $c > \psi$, and the result follows. $\square$

Motivated by the above result, we shall need the following condition in the sequel.

**Condition 2** (Realizability). $c \geq (1 + \Omega(1))\psi$, i.e $c \geq \psi(1 + \delta)$ for some $\delta > 0$, where $\psi$ is as defined in (14) and we recall that $c := \sqrt{1 - \gamma^2}$. For most of our analysis, we can work under the weaker condition $c > \psi$.

For example, the condition is always satisfied for $\alpha \to 0^+$ corresponding to sequence length $L$ which is sub-exponential (e.g., polynomial) in the dimension $d$, because we have $\psi \to 0^+$ in this case and so $\psi \leq (1 - \delta)c$ trivially, for any $\delta > 0$ and sufficiently large $d$.

## 5 Learning and Optimization

From Proposition 2, we know that under Condition 2, the oracle model parameters $(u_*, v_*)$ solve the single-locator problem. But, *can numerical optimization actually find it?* As mentioned in the introduction, unlike the case of pointwise attention considered in [Marion et al., 2025], the analysis is complicated by the softmax (2) which characterizes genuine attention layers [Vaswani et al., 2017] used in practice.

### 5.1 Preliminaries

**Projected Gradient-Descent/Gradient-Flow.** For simplicity of analysis, we shall consider a learner who has infinite samples, and therefore can directly optimize the population risk $R$ over the parametrized family (2). We shall study the dynamics of the following projected gradient descent (PGD) scheme

$$\tilde{u}_{k+1} := u_k - s P_{u_k}^\perp \nabla_u R(u, v)|_{(u,v)=(u_k, v_k)}, \quad \tilde{v}_{k+1} := v_k - s P_{v_k}^\perp \nabla_v R(u, v)|_{(u,v)=(u_k, v_k)}, \quad (28)$$

$$u_{k+1} = \tilde{u}_{k+1}/\|\tilde{u}_{k+1}\|, \quad v_{k+1} = \tilde{v}_{k+1}/\|\tilde{v}_{k+1}\|, \quad (29)$$

where $s > 0$ is the step-size and $P_u^\perp := I_d - uu^\top$ is the orthogonal projector onto the tangent space to unit-sphere $S_{d-1}$ at the point $u$. The projection step on the second line ensures that the iterates $(u_{k+1}, v_{k+1})$ remain on the the manifold $S_{d-1}^2$. For sufficiently small step size $s$, the dynamics (29) are captured by projected gradient-flow (PGF) on the manifold $S_{d-1}^2$ given by

$$\dot{u}(t) = -P_{u(t)}^\perp R(u, v)|_{(u,v)=(u(t), v(t))}, \quad \dot{v}(t) = -P_{v(t)}^\perp \nabla_v R(u, v)|_{(u,v)=(u(t), v(t))}. \quad (30)$$

The dynamics (29) and (30) induce a corresponding evolution equation for the order parameters $(\mu, \nu, \eta, \zeta, \rho)$ which will be our main object of study.

Some of our results will concern the following submanifold $\mathcal{M} \subseteq S_{d-1}^2$

$$\mathcal{M} := \{(u, v) \in S_{d-1}^2 \mid u^\top v = u^\top v_* = v^\top u_* = 0\}, \tag{31}$$

introduced by [Marion et al., 2025]. It is clear that $\mathcal{M}$ contains the oracle parameters $(u_*, v_*)$. On this submanifold the dynamics reduce to just $\mu$ and $\nu$, a two-dimensional system.

**Condition 3** (Sub-exponential block length). *In this section, we shall shall work in the frozen regime $r = 0$, corresponding to the case where $\beta_{crit} \to 0^+$, that is the number of blocks $L$ is sub-exponential in the input-dimension $d$, i.e $\log L = o(d)$. This condition is very mild, and subsumes the regime $L = o(d)$ considered in [Marion et al., 2025] as a special case.*

Since $\rho := u^\top v$ only enters the picture via the product $\rho r$, in this regime the effect $\rho$ is permanently lost; this variable disappears from the picture under the above condition.

## 5.2 Analysis of Optimization Dynamics: Arbitrary Initialization

**Proposition 3.** *For any $u, v \in S_{d-1}$, define $\mu := u^\top u_*$, $\nu := v^\top v_*$, $\eta := u^\top v_*$, $\zeta := v^\top u_*$, $\rho := u^\top v$. If $\rho = 0$, then in the limit (9), we have the following:*

$$w^\top \nabla_u R(u, v) \simeq T_1(\mu, \nu, \zeta) w^\top u_*, \ w^\top \nabla_v R(u, v) \simeq T_3(\mu, \nu, \zeta) w^\top u_* + T_2(\mu, \nu, \zeta) w^\top v_*, \tag{32}$$

*uniformly on $w \in S_{d-1}$, where the functions $T_1, T_2, T_2 : [-1, 1]^3 \to \mathbb{R}$ are as given in Appendix E.1.*

Thus, (asymptotically) the gradients of the risk $R$ are trapped in the 2-dimensional subspace of $\mathbb{R}^d$ spanned by the oracle parameters $(u_*, v_*)$.

**Corollary 2.** *For sufficiently small step-size in the iteration scheme (29), the equations of motion for overlaps $(\mu, \nu, \eta, \zeta)$ are given by the following 4-dimensional gradient-flow:*

$$\dot{\mu}(t) = F_1(\mu(t), \nu(t), \zeta(t)), \quad \dot{\nu}(t) = F_2(\mu(t), \nu(t), \zeta(t)), \tag{33}$$

$$\dot{\eta}(t) = F_3(\mu(t), \nu(t), \eta(t), \zeta(t)), \quad \dot{\zeta}(t) = F_4(\mu(t), \nu(t), \zeta(t)). \tag{34}$$

**Stronger Results in the Case of Linear Link Function.** Even in the greatly simplified case of point-wise / non-softmax attention (1) considered in the reference work [Marion et al., 2025], their analysis was further restricted to the setting where the optimization scheme (29) is initialized on the sub-manifold $\mathcal{M}$. This is problematic because by definition, choosing a point on $\mathcal{M}$ presupposes knowledge of the oracle parameter $(u_*, v_*)$. Our next result closes this gap and shows that this sub-manifold indeed contains all the stationary points, and is therefore eventually attained by the dynamics (29), irrespective of the initialization.

**Proposition 4.** *For the linear link function $\sigma(t) := t$, the sub-manifold $\mathcal{M}$ contains all the stationary points of the 4-dimensional dynamics (30). In fact, the only stationary points are $(\pm 1, \pm 1, 0, 0)$, of which $(1, 1, 0, 0)$ (corresponding to the oracle model parameters $(u_*, v_*)$) is the only stable one (more precisely, it is an attractor/sink); the others are saddles and sources.*

## 5.3 Dynamics on the Submanifold $\mathcal{M}$

We now consider the dynamics (29) in the regime where it has entered the submanifold $\mathcal{M}$ defined in (31) (e.g., via initialization), and show a drastic simplification of the picture.

**Proposition 5.** *The 4-dimensional dynamics (29) fixes the submanifold $\mathcal{M}$. That is, once the dynamics (29) enters $\mathcal{M}$, it remains there.*

**Proposition 6.** *In the limit of vanishing step-size, the equations of motion of the overlap variables $\mu$ and $\nu$, induced (30) are given by the following 2-dimensional gradient-flow:*

$$\dot{\mu}(t) = F_1^0(\mu(t), \nu(t)), \quad \dot{\nu}(t) = F_2^0(\mu(t), \nu(t)), \tag{35}$$

*where the scalar fields $F_1^0, F_2^0 : [-1, 1]^2 \to \mathbb{R}$ are as defined in (71) and (72) respectively.*

**Stationary Points.** Let $E = \{(\mu, \nu) \subseteq [-1, 1]^2 \mid F_1^0(\mu, \nu) = F_2^0(\mu, \nu) = 0\}$ be the stationary points of the 2-dimensional dynamics dynamics (35). Consider the following subsets of $[-1, 1]^2$

$$E_1 := \{(\pm 1, \pm 1)\}, \quad E_2 := \{(\pm 1, \nu) \mid \nu \in (-1, 1), \bar{\sigma}'_\gamma(\nu) = 0\}, \tag{36}$$

$$E_3 := \{(\mu, -1) \mid \mu \in (-1, 1), p(\mu) = \bar{\sigma}_\gamma(-1)/\bar{\sigma}_\gamma(1)\}, \tag{37}$$

$$E_4 := \{(\mu, \nu) \in (-1, 1)^2 \mid \bar{\sigma}'_\gamma(\nu) = 0, p(\mu) = \bar{\sigma}_\gamma(\nu)/\bar{\sigma}_\gamma(1)\}. \tag{38}$$

**Proposition 7.** $E = E_1 \cup E_2 \cup E_3 \cup E_4$ *is a partitioning of the set of stationary points of the 2-dimensional dynamics* (35).

*Proof.* The $E$ of stationary points correspond to pairs $(\mu, \nu)$ for which $F_1^0(\mu, \nu) = F_2^0(\mu, \nu) = 0$. By definition this translates to

$$0 = (1 - \mu^2)A(\mu)B(\mu, \nu) = (1 - \mu^2)A(\mu)2(p(\mu)\bar{\sigma}_\gamma(1) - \bar{\sigma}_\gamma(\nu)), \quad 0 = -2(1 - \nu^2)p(\mu)\bar{\sigma}'_\gamma(\nu).$$

But $A(\mu) := p(\mu)(1 - p(\mu))c\beta d$ and $p(\mu)$ are always positive (since $p(\mu) \in (0, 1)$) and so they can be canceled out in the above equations to give

$$(1 - \mu^2)(p(\mu)\bar{\sigma}_\gamma(1) - \bar{\sigma}_\gamma(\nu)) = 0, \quad (1 - \nu^2)\bar{\sigma}'_\gamma(\nu) = 0.$$

By manipulating the equations, it is easy to see that the all the solutions can be organized into the sets $E_1$, $E_2$, $E_3$, and $E_4$ respectively. Each $E_k$ vanishes exactly one of the two factors in either of the two equations above (four possibilities in total). $\square$

A complete classification of the stationary points is provided in Appendix A.

### 5.4   Convergence to A Stationary Point

Our analysis would be incomplete without showing that the PGD dynamics (29) actually converges to a stationary point. The following result is proved in Appendix G.

**Proposition 8.** *Under some smoothness conditions on the link function $\sigma$ (made explicit in the appendix), the following holds: If $(u_0, v_0) \in \mathcal{M}$, then the PGD dynamics (29) converges to a stationary point of the population risk functional $R$.*

Taken together with with Proposition 4, Proposition 5, Proposition 7, alongside the results in Appendix A on the classification of stationary points, we infer the following:

- In the case of the identity link functions $\sigma(t) \equiv t$, (projected) gradient descent on the population risk function $R$ converges to the group truth model parameters $(u_*, v_*)$, irrespective of the initialization $(u_0, v_0)$. This result is much stronger that [Marion et al., 2025] which required $(u_0, v_0) \in \mathcal{M}$, even though the latter considered the much simpler case of pointwise self-attention (1).

- In the general case, we have the same convergence as above, provided the initialization $(u_0, v_0)$ is on the manifold $\mathcal{M}$.

## 6   Concluding Remarks

We present an end-to-end theories of *softmax* self-attention and an interesting task, the *single-locator regression problem* proposed in [Marion et al., 2025]. Building on the pointwise/non-softmax analysis of Marion et al. [2025], we give closed-form formulas for the population risk and gradient flow, pinpointing why attention solves tasks that defeat linear models. Our proof unites tools from statistical physics, mean-field theory, and Riemannian optimization, while eliminating the delicate initialization assumptions of previous work. Thus a ubiquitous yet opaque layer becomes one whose behavior can now be predicted, analyzed, and engineered.

**Limitations and Future Directions.** A nature next step would be to extend our proposed theory to (1) empirical risk minimization (i.e finite-sample analysis) and (2) *multi-locator* regression problem, where instead of a single secret token index [Marion et al., 2025], several secret indices must be recovered. The latter would encompass tasks such as the well-known sparse parity problem, opening the door to richer combinatorial analyses of attention. With some extra effort, the core ideas technical ideas developed in our work would directly extend to the this setting.

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

# Appendix

## Contents

## A  Classification of Stationary Points: Sinks, Sources, Saddles

We start with the following general result which shows that oracle model with parameters $(u_*, v_*)$ corresponding to $\mu = \nu = 1$ is a stable stable stationary point of the dynamics (35), even in the case of nonlinear link function $\sigma$.

**Proposition 9.** *The point* $(1, 1)$ *(corresponding to the oracle model parameters* $(u_*, v_*)$*) is always an attractor/sink of the 2-dimensional dynamics* (35)*.*

As with all the other results in this section, the proof of the above result is provided in Section F.

We now give a complete classification of the stationary points $E = \cup_k E_k$ for various choices of the link function $\sigma$.

**Proposition 10.** *We have the following classification of the stationary points of the 2-dimensional dynamics* (35) *induced by different choice of the link function $\sigma$.*

*(A) **Linear link function.** For $\sigma(t) := t$, the stationary points are: $(\pm 1, \pm 1)$, of which $(1, 1)$ is a sink (stable), $(1, -1)$ is a source (unstable), and $(-1, \pm 1)$ are saddles (unstable).*

*(B) **Quadratic link function.** For $\sigma(t) := t^2$, the stationary points are: $(\pm 1, \pm 1)$, $(\pm 1, 0)$, and $(\psi, 0)$, where $\psi$ is the thermodynamic parameter introduced in* (14). *Moreover, $(1, \pm 1)$ are sinks (stable); $(-1, \pm 1)$, $(-1, 0)$, and $(\psi, 0)$ are saddles (unstable); $(1, 0)$ is a source (unstable).*

*Note that because of the evenness of this link function, the stable stationary points $(1, \pm 1)$ both correspond to the oracle parameters $(u_*, v_*)$.*

*(C) **ReLU link function.** Consider the link function $\sigma(t) := \max(t, 0)$, and suppose $\gamma \in [1/2, 1)$. Then, the dynamics has 4 stationary points: $(\pm 1, \pm 1)$, of which $(1, 1)$ is a sink (stable), $(1, -1)$ is a saddle (unstable), and $(\pm 1, -1)$ are degenerate.*

Thus, for all these link functions, the iteration scheme (29) with sufficiently small step-size is guaranteed to converge to the oracle parameters $(u_*, v_*)$.

# B   Misc: Additional Theoretical Results

**Proposition 11.** *For any $\nu, \zeta, \rho \in [-1, 1]$, set $a := c\zeta\sqrt{d}$, $h := a/\gamma$, $b := \rho r\sqrt{d}$, $b_0 := \max(b, 0)$, and $q := a\Phi(h) + \gamma\varphi(h)$. For different choices of the link function $\sigma$, the function dual function $\bar\sigma_\gamma$, and the $H_k$'s in Proposition 1 have the following closed form expressions for any $\nu, \zeta, \rho \in [-1, 1]$.*

*(A) **Linear link function.** If $\sigma(t) := t$, then $\bar\sigma_\gamma(\nu) = \gamma^2\nu$, and*

$$H_1(\nu, \zeta, \rho) = \bar\sigma_\gamma(\nu) + (a - b)^2, \quad H_2(\nu, \zeta, \nu) = \bar\sigma_\gamma(\nu) - (a - b)b. \tag{39}$$

*(B) **Quadratic link function.** If $\sigma(t) := t^2$, then $\bar\sigma_\gamma(\nu) = \gamma^4(1 + 2\nu^2)$, and*

$$H_1(\nu, \zeta, \rho) = \bar\sigma_\gamma(\nu) + a^4 + 2a^2\gamma^2(1 + 2\nu) - 2b^2(a^2 + \gamma^2) + b^4, \tag{40}$$

$$H_2(\nu, \zeta, \rho) = \bar\sigma_\gamma(\nu) + \gamma^2 a^2 - b^2(a^2 + 2\gamma^2) + b^4. \tag{41}$$

*(C) **ReLU link function.** If $\sigma(t) := \max(t, 0)$, then $\bar\sigma_\gamma(\nu) = (\sqrt{1 - \nu^2} + \nu\arccos(-\nu))\gamma^2/(2\pi)$, the well-known "arc-cosine" kernel, and*

$$H_1(\nu, \zeta, \rho) = \bar\sigma_\gamma(\nu) + \gamma^2 h^2\Phi_2(h, h; \nu) + 2\gamma^2 h\varphi(h)\Phi(h\frac{\sqrt{1 - \nu}}{\sqrt{1 + \nu}}) - 2qb_0 + b_0^2, \tag{42}$$

$$H_2(\nu, \zeta, \rho) = \bar\sigma_\gamma(\nu) + \gamma^2 h\varphi(h)\Phi(h\frac{\sqrt{1 - \nu}}{\sqrt{1 + \nu}}) - (q + \frac{\gamma}{\sqrt{2\pi}})b_0 + b_0^2, \tag{43}$$

*where $\varphi$ is the standard Gaussian pdf, $\Phi$ is the corresponding cdf, and $\Phi_2(\cdot, \cdot; \nu)$ is the cdf of the standard bi-variate Gaussian with unit variance and correlation coefficient $\nu$.*

The proof for the case linear and quadratic link functions reveals that the results can be readily generalized to general powers by making use of hyper-geometric functions of type $_2F_1$.

# C   Relevant Statistical Physics for Machine Learning

## C.1   High-dimensional Analysis of Abstract Nadaraya-Watson Estimator (Local Learning)

Fix unit-vectors $u, v \in \mathbb{R}^N$. Consider a size-$n$ random energy model (REM) [Derrida, 1981] in $N$ dimensions, with energy levels $E_1, \ldots, E_n$, where $E_i := \sqrt{N}x_i^\top u$, i.e independent energy levels from $\mathcal{N}(0, N)$, and consider the sum

$$g := \sum_{i=1}^n p_i\sigma(x_i^\top v/\sqrt{N}), \text{ with } p_i := \frac{e^{\beta E_i}}{Z}, Z := \sum_{j=1}^n e^{\beta E_j},$$

for some link function $\sigma \in L^2(\mathcal{N}(0, 1))$, which **doesn't depend** on $N$. We seek a deterministic equivalent in the limit

$$N, n \to \infty, \quad \frac{\log n}{N} \to \alpha. \tag{44}$$

The following result is an adaptation of the main result of Zavatone-Veth and Pehlevan [2025] to the case of Gaussian covariates.

**Proposition 12.** *In the limit* (44)*, it holds a.s that*

$$\frac{\log Z}{\beta N} \to \psi, \quad g \to \sigma(\rho r), \tag{45}$$

$$with \ \rho := u^\top v, \quad \psi = \phi/\beta, \quad \phi = \alpha + \beta r - r^2/2, \tag{46}$$

$$r = \min(\beta, \beta_{crit}), \quad \beta_{crit} = \sqrt{2\alpha}. \tag{47}$$

*Proof.* We use Laplace method of integration coupled with the recipe of Lucibello and Mézard [2024], Zavatone-Veth and Pehlevan [2025]. Thus, we write

$$g \to \frac{\int\int e^{\phi(t,q)N} \sigma(q)\,\mathrm{d}t\mathrm{d}q}{\int\int e^{\phi(t,q)N}\,\mathrm{d}t\mathrm{d}q}, \tag{48}$$

where we have introduced overlaps

$$t = E_i/\sqrt{N} = x_i^\top u/\sqrt{N}, \quad q = x_i^\top v/\sqrt{N}.$$

The potential $\phi$ has an energetic part $\beta t$ and an entropic part $\alpha - s(t, q)$. In order to determine the function $s$, we use large-deviation methods. Consider the random bi-variate Gaussian random vector $z = (x_i^\top u, x_i^\top v)$, with covariance matrix $N\Sigma$, where

$$\Sigma = \begin{bmatrix} 1 & \rho \\ \rho & 1 \end{bmatrix}, \quad \Sigma^{-1} = \frac{1}{\kappa} \begin{bmatrix} 1 & -\rho \\ -\rho & 1 \end{bmatrix}, \quad \text{with } \rho := u^\top v, \quad \kappa := 1 - \rho^2. \tag{49}$$

The (normalized) log-MGF of $w$ is given by

$$\frac{1}{N} \log \mathbb{E}\, e^{\hat{c}^\top z} = \frac{1}{N} \hat{c}^\top N\Sigma\hat{c}/2 = \hat{c}^\top \Sigma\hat{c}/2 =: \zeta(\hat{c}), \text{ for any } \hat{c} \in \mathbb{R}^2. \tag{50}$$

We take $s$ to be the Legendre transform of $\zeta$, i.e

$$s(c) = \sup_{\hat{c} \in \mathbb{R}^2} \hat{c}^\top c - \zeta(\hat{c}) = c^\top \Sigma^{-1} c/2 = t^2/(2\kappa) - \rho t q/\kappa + q^2/(2\kappa)$$

$$= (t - \rho q)^2/(2\kappa) + (1 - \rho^2)q^2/(2\kappa) = (t - \rho q)^2/(2\kappa) + q^2/2,$$

for any $c = (t, q)$. The condensation region the corresponds to $s(c) \geq \alpha$, i.e $c^\top \Sigma c/2 \geq \alpha$, which is an ellipsoid in $c$-space. Consider the re-parametrization in polar coordinates

$$t - \rho q = r\sqrt{1 - \rho^2}\cos\theta, \quad q = r\sin\theta. \tag{51}$$

That is, $q = r\sin\theta$ and $t = r\sqrt{1 - \rho^2}\cos\theta + \rho r\sin\theta$. This gives $s(r, \theta) = r^2/2$, and our potential takes the form

$$\Phi(r, \theta) = \beta t(r, \theta) + \alpha - s(r, \theta) = \alpha + \beta r \cdot \left(\sqrt{1 - \rho^2}\cos\theta + \rho\sin\theta\right) - r^2/2, \tag{52}$$

and we must now maximize w.r.t $(r, \theta)$ outside the condensation region. Maximizing $\Phi(r, \theta)$ w.r.t $\theta$ gives

$$\theta = \arctan(\rho/\sqrt{1 - \rho^2}).$$

This in turn gives $\cos\theta = \sqrt{1 - \rho^2}$ and $\sin\theta = \rho$. Plugging into $\phi$ gives

$$\Phi = \alpha + \beta r - r^2/2, \quad q = \rho r.$$

Moreover, the condensation region is now given by $\alpha \leq s(r, \theta) = r^2/2$, i.e $r \geq \sqrt{2\alpha}$. Outside of this region (i.e for $r < \sqrt{2\alpha}$), maximizing $\Phi$ w.r.t $r$ gives $r = \beta$.

Putting things together, we get the following final form for our potential

$$\phi = \phi(\alpha, \beta) = \alpha + \beta r - r^2/2, \quad \text{with } r = \min(\beta, \beta_{crit}), \quad \beta_{crit} = \sqrt{2\alpha}.$$

It is a standard result for the REM that $(\beta N)^{-1} \log Z \to \psi = \phi/\beta$. Finally, plugging everything into the RHS of (48) then gives $g \simeq (e^{\phi N}/Z)\sigma(\rho r) \simeq (e^{\phi N}/e^{\phi N})\sigma(\rho r) = \sigma(\rho r)$, as claimed. $\qquad\square$

For our purposes, we would like to compute sums of the form $g = \sum_i p_i \sigma(x_i^\top v)$ and not $\sum_i p_i \sigma(x_i^\top v/\sqrt{N})$ as in Proposition 12. To address this, we can't simply consider a new function $h_N(t) := \sigma(t\sqrt{N})$ to write $f = \sum_i h_N(x_i^\top v/\sqrt{N})$ and then apply Proposition 12. The issues is that $h_N$ is not fixed in $L^2(\mathcal{N}(0,1))$ but itself varies with the dimension $N$ which is tending to infinity. In the special case when $\sigma$ is positive-homogeneous, we can effectively factor out this dimension-dependence and correctly apply Proposition 12. Viz,

**Corollary 3.** *Let $(p_i)_i$ be the Gibbs distribution from Proposition 12. If $\sigma \in L^2(\mathcal{N}(0,1))$ is positively-homogeneous, then in the limit* (44)*, it holds that*

$$\sum_{i=1}^{n} p_i \sigma(x_i^\top v) \simeq \sigma(\rho r \sqrt{N}). \tag{53}$$

*Proof.* Indeed, positive-homogeneity means that there exists $m > 0$ such that $\sigma(ut) = u^m \sigma(t)$ for all $u > 0$ and $t \in \mathbb{R}$. In particular, we have $\sigma(x_i^\top v/\sqrt{N}) = \sigma(x_i^\top v)/N^{m/2}$, i.e. $\sigma(x_i^\top v) = N^{m/2}\sigma(x_i^\top v/\sqrt{N})$. The result then follows from Proposition 12:

$$\sum_{i=1}^{n} p_i \sigma(x_i^\top v) = N^{m/2} \sum_{i=1}^{n} p_i \sigma(x_i^\top v/\sqrt{N}) \simeq N^{m/2}\sigma(\rho r) = \sigma(\rho r \sqrt{N}).$$

$\square$

## C.2 A Novel Extension of the Nadaraya-Watson Estimator

We now extend Proposition to the case of multivariate link functions. A slight modification of our arguments from Gaussian to spherical data immediately gives a non-trivial extension of the main result in [Zavatone-Veth and Pehlevan, 2025].

Let $u, v_1, \ldots, v_k \in \mathbb{R}^N$ be unit-vectors, and let $\rho_j := u^\top v_j$ be the cosine of the angle that $u$ makes with each $v_j$. Let $x_1, \ldots, x_n$ be iid from $\mathcal{N}(0, I_N)$, and consider the following generalized NW estimator

$$g := \sum_{i=1}^{n} \frac{e^{\beta\sqrt{N}x_i^\top u}}{Z} F(x_i^\top v_1/\sqrt{N}, \ldots, x_i^\top v_k/\sqrt{N}), \text{ with } Z := \sum_{i=1}^{n} e^{\beta\sqrt{N}x_i^\top u}, \tag{54}$$

where the $F$ is an $L^2(\mathcal{N}(0, I_k))$ function and $k \geq 1$ is a fixed integer. We seek a deterministic equivalent for $g$ in the limit (44).

**Proposition 13.** *In the limit* (44)*, it holds that $g \to F(\rho r_1, \ldots, \rho r_k)$ a.s.*

*Proof.* Once again, we will follow the line of thought of Lucibello and Mézard [2024]. The Laplace method let's us write:

$$g \simeq \frac{\int \int \ldots \int e^{\phi(t,q_1,\ldots,q_k)N} F(q_1, \ldots, q_k) \, dt \, dq_1 \ldots dq_k}{\int \int \ldots \int e^{\phi(t,q_1,\ldots,q_k)N} \, dt \, dq_1 \ldots dq_k}, \tag{55}$$

with the definition of overlaps

$$t \leftarrow x_i^\top u/\sqrt{N}, \quad q_1 \leftarrow x_i^\top v_1/\sqrt{N}, \quad \ldots, \quad q_k \leftarrow x_i^\top v_k/\sqrt{N}. \tag{56}$$

Now, we need to compute the log-MGF of the Gaussian random vector $z := (x_i^\top u, x_i^\top v_1, \ldots, x_i^\top v_k)$. Its covariance matrix is $N\Sigma$, where $\Sigma \in \mathbb{R}^{(k+1)\times(k+1)}$ is the covariance matrix of $z/\sqrt{N}$, given by

$$\Sigma = \begin{bmatrix} 1 & \rho^\top \\ \rho & VV^\top \end{bmatrix}, \text{ with } \rho = \begin{bmatrix} \rho_1 \\ \vdots \\ \rho_k \end{bmatrix} \in \mathbb{R}^{k\times 1}, \text{ and } V := \begin{bmatrix} v_1 \\ \vdots \\ v_k \end{bmatrix} \in \mathbb{R}^{k\times N}. \tag{57}$$

We thus compute the normalized log-MGF of $z$ as

$$\zeta(\hat{c}) = \frac{1}{N} \log \mathbb{E}\, e^{\hat{c}^\top z} = \frac{1}{N}\hat{c}^\top N\Sigma\hat{c}/2 = \hat{c}^\top \Sigma \hat{c}/2, \text{ for any } \hat{c} \in \mathbb{R}^{k+1}.$$

The Legendre transform of $\zeta$ is of course given by

$$s(c) = \sup_{\hat{c} \in \mathbb{R}^{k+1}} \hat{c}^\top c - \zeta(\hat{c}) = c^\top \Sigma^{-1} c/2, \text{ for any } c = (t, q_1, \ldots, q_k) \in \mathbb{R}^{k+1}.$$

Recall that the condensation region in $c$-space is then the exterior of the set by $s(c) \le \alpha$, i.e the ellipsoid $\mathcal{E} \subseteq \mathbb{R}^{k+1}$ given by

$$c^\top \Sigma^{-1} c/2 \le \alpha,$$

Now, for any fixed value $r^2/2$ of $s(c)$, the sought-for potential $\phi$ has an energetic part $\beta t$ and an entropic part $\alpha - s(c) = \alpha - r^2/2$, i.e has the form

$$\phi = \beta t + \alpha - r^2/2.$$

We now maximize $\phi$ subject to the constraint $s(c) = r^2/2$, i.e $c^\top \Sigma^{-1} c = r^2$. This is a linear maximization problem (since $t = c^\top e_1$) with quadratic constraint. The method of Lagrange multipliers gives $c \propto \Sigma e_1$, i.e $c$ is proportional to the first row of $\Sigma$ which is the vector $(1, \rho_1, \ldots, \rho_k)$. Plugging the constraint $c^\top \Sigma^{-1} c = r^2$ gives $c = r \Sigma e_1 = (r, \rho r_1, \ldots, \rho r_k)$. The potential then takes the form

$$\phi = \alpha + \beta r - r^2/2, \quad t = c^\top e_1 = r, \quad q_j = \rho r_j, \forall j. \tag{58}$$

The condensation region $\mathcal{E}'$ is then $r \ge \sqrt{2\alpha} =: \beta_{crit}$. In this region, maximizing $\phi$ w.r.t $r$ gives $r = \beta$. Thus, we must have $r = \min(\beta, \beta_{crit})$. Combining with (55), we deduce that

$$g \simeq \frac{\int \int \ldots \int e^{\phi(t, q_1, \ldots, q_k)N} F(\rho r_1, \ldots, \rho r_k) \, dt \, dq_1 \ldots dq_k}{\int \int \ldots \int e^{\phi(t, q_1, \ldots, q_k)N} \, dt \, dq_1 \ldots dq_k} \simeq \frac{e^{\phi N} F(\rho r_1, \ldots, \rho r_k)}{e^{\phi N}}$$

$$= F(\rho r_1, \ldots, \rho r_k),$$

as claimed. $\qquad \square$

We have the following generalization of Corollary 4.

**Corollary 4.** *Let $F$ be as in Proposition 13. If in addition $F$ is positive-homogeneous w.r.t to each input (the orders of the homogeneity are allowed to be different), then in the limit (44), it holds a.s that*

$$\sum_i p_i F(x_i^\top v_1, \ldots, x_i^\top v_k) \simeq F(r\rho_1 \sqrt{N}, \ldots, r\rho_k \sqrt{N}). \tag{59}$$

Recall that $F$ being positive-homogeneity of order $(m_1, \ldots, m_k) \in \mathbb{R}_+^k$ means that

$$F(u_1 t_1, \ldots, u_k t_k) = F(t_1, \ldots, t_k) \prod_{j=1}^k u_j^{m_j},$$

for any $t_1, \ldots, t_k \in \mathbb{R}$ and $u_1, \ldots, u_k > 0$.

*Proof of Corollary 4.* Indeed, taking $u_j \equiv 1/\sqrt{N}$ and $t_j \equiv x_i^\top v_j$ gives

$$F(x_i^\top v_1, \ldots, x_i^\top v_k) N^{-\sum_j m_j/2} = F(x_i^\top v_1/\sqrt{N}, \ldots, x_i^\top v_k/\sqrt{N}).$$

Thanks to Proposition 13, we know that RHS $\simeq F(r\rho_1, \ldots, r\rho_k)$. We deduce that

$$F(x_i^\top v_1, \ldots, x_n^\top v_k) \simeq N^{\sum_j m_j/2} F(r\rho_1, \ldots, r\rho_k) = F(r\rho_1 \sqrt{N}, \ldots, r\rho_k \sqrt{N}),$$

as claimed. $\qquad \square$

# D High-Dimensional Representation of Estimator and Its Risk

## D.1 Case of Linear Link Function: Proof of Corollary 1

We start with an instructive self-contained proof of Corollary 1. Following the decomposition (11), we only need to estimate $f_2 = \sum_{\ell \neq \ell_*} q_\ell x_\ell^\top v$, where $q_\ell := e^{\beta E_\ell}/Z_{-\ell_*}$, $E_\ell := \sqrt{d} x_\ell^\top u$, $Z := \sum_{\ell \neq \ell_*} e^{\beta E_\ell}$. We can decompose $x_\ell^\top v = \rho x_\ell^\top u + \sqrt{1-\rho^2} z_\ell$, where the $z_\ell$'s are $\mathcal{N}(0,1)$, and independent of $Xu$. This gives

$$f_2 = \rho \sum_{\ell \neq \ell_*} q_\ell x_\ell^\top u + \sqrt{1-\rho^2} \sum_{\ell \neq \ell_*} q_\ell z_\ell.$$

Conditioned on $Xu$, the second sum is a centered Gaussian distribution with variance equal to $(1-\rho^2) \sum_{\ell \neq \ell_*} q_\ell^2 \leq 1 - \rho^2 \leq 1$. For the first sum, observe that

$$\frac{1}{\sqrt{d}} \sum_{\ell \neq \ell_*} q_\ell x_\ell^\top u = \frac{1}{d} \sum_{\ell \neq \ell_*} \frac{e^{\beta E_i}}{Z_{-\ell_*}} E_i = \frac{1}{d} \frac{\partial \log Z_{-\ell_*}}{\partial \beta} \to \frac{\partial \phi}{\partial \beta} = r, \tag{60}$$

and the result follows. The step "$\to$" is a classical result in the analysis of the REM, while the last step "$\partial_\beta \phi = r$ follows from the definition of $\phi$ and $r$ in (14) and (17). We deduce that $f_2 \simeq \rho r \sqrt{d}$, and so $f(X; u, v) \simeq p x_{\ell_*}^\top v + (1-p) \rho r \sqrt{d}$ as claimed. The claimed formula for the population risk $R(u, v)$ is then obtained by plugging the previous formula for $f$ into definition (10), and then computing basic Gaussian integrals. $\qquad\square$

## D.2 Proof of Proposition 1

Indeed, following the decomposition (11), we know that $f \simeq p\sigma(x_{\ell_*}^\top v) + (1-p)f_2$, and we only need to estimate $f_2 = \sum_{\ell \neq \ell_*} q_\ell \sigma(x_\ell^\top v)$, where $q_\ell := e^{\beta E_\ell}/Z_{-\ell_*}$, $E_\ell := \sqrt{d} x_\ell^\top u$, $Z := \sum_{\ell \neq \ell_*} e^{\beta E_\ell}$, and $p = p(u^\top u_*)$ is as defined in formula (20). Corollary 3 with $N = d$ and $n = L - 1$ gives

$$f_2 \simeq \sigma(\rho r \sqrt{d}).$$

We deduce that $f \simeq p\sigma(x_{\ell_*}^\top v) + (1-p)\sigma(\rho r \sqrt{d})$ as claimed.

The claimed formula for the risk $R(u, v)$ is then a matter of direct Gaussian integration. $\qquad\square$

# E Equations of Motion

## E.1 Auxiliary Functions

Define auxiliary functions $A : [-1, 1] \to \mathbb{R}$, and $B, T_1, T_2, T_3, F_1, F_2, F_4 : [-1, 1]^3 \to \mathbb{R}$, and $F_3 : [-1, 1]^4 \to \mathbb{R}$ by

$$A(\mu) := p(\mu)(1 - p(\mu))c\beta d > 0, \tag{61}$$

$$B(\mu, \nu, \zeta) := 2(p(\mu)h_1(1, \zeta) - h_2(\nu, \zeta)), \tag{62}$$

$$T_1(\mu, \nu, \zeta) := A(\mu)B(\mu, \nu, \zeta), \tag{63}$$

$$T_2(\mu, \nu, \zeta) := -2p(\mu)\partial_\nu h_2(\nu, \zeta), \tag{64}$$

$$T_3(\mu, \nu, \zeta) := p(\mu)\partial_\zeta[h_1(1, \zeta) - 2h_2(\nu, \zeta)], \tag{65}$$

$$F_1(\mu, \nu, \zeta) := -(1 - \mu^2)T_1(\mu, \nu, \zeta), \tag{66}$$

$$F_2(\mu, \nu, \zeta) := \zeta\nu T_3(\mu, \nu, \zeta) - (1 - \nu^2)T_2(\mu, \nu, \zeta), \tag{67}$$

$$F_3(\mu, \nu, \eta, \zeta) := \eta\mu T_1(\mu, \nu, \zeta), \tag{68}$$

$$F_4(\mu, \nu, \zeta) := -(1 - \zeta^2)T_3(\mu, \nu, \zeta) - \zeta\nu T_2(\mu, \nu, \zeta), \tag{69}$$

where $h_k(\nu, \zeta) := H_k(\nu, \zeta, 0)$, and the $H_k$ are as defined in (23). In particular, when $\eta = \zeta = 0$ (as on the sub-manifold $\mathcal{M}$), the above formulae drastically reduce to

$$B(\mu, \nu, 0) = 2(p(\mu)\bar{\sigma}_\gamma(1) - \bar{\sigma}_\gamma(\nu)), \quad T_1(\mu, \nu, 0) = A(\mu)B(\mu, \nu, 0), \tag{70}$$

$$F_1(\mu, \nu, 0) = F_1^0(\mu, \nu) := -(1 - \mu^2)T_1(\mu, \nu, 0), \tag{71}$$

$$T_2(\mu, \nu, 0) := -2p(\mu)\bar{\sigma}_\gamma'(\nu), \quad F_2(\mu, \nu, 0) = F_2^0(\mu, \nu) := -(1 - \nu^2)T_2(\mu, \nu, 0). \tag{72}$$

### E.2 Proof of Proposition 3 (High-Dimensional Representation of Gradient of Population Risk)

Recall $R(u, v) = \mathbb{E}[\delta(X; u, v)^2]$, where $\delta(X; u, v) := f(X; u, v) - \sigma(x_{\ell_*}^\top v_*)$. Differentiating w.r.t $u$ and $v$ gives

$$\nabla_u R(u, v) = \mathbb{E}[\delta(X; u, v)\nabla_u f(X; u, v)], \quad \nabla_v R(u, v) = \mathbb{E}[\delta(X; u, v)\nabla_v f(X; u, v)] \quad (73)$$

We already know that

$$\delta(X; u, v) \simeq p\sigma(x_{\ell_*}^\top v) + (1-p)\sigma(\rho r\sqrt{d}) - \sigma(x_{\ell_*}^\top v_*).$$

We now need to estimate $\nabla_u f(X; u, v)$ and $\nabla_v f(X; u, v)$. Setting $\lambda := \beta\sqrt{d}$, one computes

$$\nabla_v f = \sum_\ell \frac{e^{\lambda x_i^\top u}}{Z}\sigma'(x_\ell^\top v)x_\ell = p\sigma'(x_{\ell_*}^\top v)x_{\ell_*} + (1-p)\sum_{\ell \neq \ell_*} \frac{e^{\lambda x_i^\top u}}{Z}\sigma'(x_\ell^\top v)x_\ell,$$

$$\frac{1}{\lambda}\nabla_u f = \frac{1}{\lambda}\sum_\ell \nabla_u \frac{e^{\lambda x_i^\top u}}{Z}\sigma(x_\ell^\top v)$$

$$= \sum_\ell \frac{e^{\lambda x_\ell^\top u}}{Z}\sigma(x_\ell^\top v)x_\ell - \sum_\ell \frac{e^{\lambda x_i^\top u}}{Z}\sigma(x_\ell^\top v) \cdot \frac{1}{Z^2}\sum_\ell e^{\lambda x_\ell^\top u}x_\ell$$

$$= \sum_\ell \frac{e^{\lambda x_i^\top u}}{Z}\sigma(x_\ell^\top v)x_\ell - f\sum_\ell \frac{e^{\lambda x_i^\top u}}{Z}x_\ell$$

$$\simeq p \cdot (\sigma(x_{\ell_*}^\top v) - f)x_{\ell_*} + (1-p)\sum_{\ell \neq \ell_*} \frac{e^{\lambda x_\ell^\top u}}{Z_{-\ell_*}}\sigma(x_\ell^\top v)x_\ell - f \cdot (1-p)\sum_{\ell \neq \ell_*} \frac{e^{\lambda x_\ell^\top u}}{Z_{-\ell_*}}x_\ell,$$

where $p = p(\mu) := 1/(1 + e^{-(c\mu - \psi)\beta d})$ and $\mu := u^\top u_*$ as usual.

Now, for any $w \in S_{d-1}$, Corollary 3 and Corollary 4 (with $N = d$ and $n = L - 1$) give

$$f \simeq p\sigma(x_{\ell_*}^\top v) + (1-p)\sigma(\rho r\sqrt{d}), \tag{74}$$

$$\delta \simeq p(\sigma(x_{\ell_*}^\top v) - \sigma(\rho r\sqrt{d})) - (\sigma(x_{\ell_*}^\top v_*) - \sigma(\rho r\sqrt{d}))$$

$$= p\sigma_{\gamma,\zeta,\rho}(z^\top v) - \sigma_{\gamma,0,\rho}(z^\top v_*), \tag{75}$$

$$\sigma(x_{\ell_*}^\top v) - f \simeq (1-p)\sigma(x_{\ell_*}^\top v) - (1-p)\sigma(\rho r\sqrt{d})$$

$$= (1-p)(\sigma(x_{\ell_*}^\top v) - \sigma(\rho r\sqrt{d})) = (1-p)\sigma_{\gamma,\zeta,\rho}(z^\top v), \tag{76}$$

$$\sum_{\ell \neq \ell_*} \frac{e^{\lambda x_\ell^\top u}}{Z_{-\ell_*}}\sigma(x_\ell^\top v)x_\ell^\top w \simeq \sigma(\rho r\sqrt{d})r\sqrt{d}u^\top w, \tag{77}$$

$$\sum_{\ell \neq \ell_*} \frac{e^{\lambda x_\ell^\top u}}{Z_{-\ell_*}}x_\ell^\top w \simeq r\sqrt{d}u^\top w, \tag{78}$$

$$\sum_{\ell \neq \ell_*} \frac{e^{\lambda x_i^\top u}}{Z}\sigma'(x_\ell^\top v)x_\ell^\top w \simeq \sigma'(\rho r\sqrt{d})r\sqrt{d}u^\top w. \tag{79}$$

From the above display, we deduce that

$$w^\top \nabla_v f \simeq p\sigma'(x_{\ell_*}^\top v)x_{\ell_*}^\top w + (1-p)\sigma'(\rho r\sqrt{d})r\sqrt{d}u^\top w$$

$$= p\nabla_v\sigma(x_{\ell_*}^\top v)^\top w + (1-p)\nabla_v\sigma(\rho r\sqrt{d})^\top w$$

$$= p\nabla_v(\sigma(x_{\ell_*}^\top v) - \sigma(\rho r\sqrt{d}))^\top w + \nabla_v\sigma(\rho r\sqrt{d})^\top w$$

$$= p\nabla_v\sigma_{\gamma,\zeta,\rho}(z^\top v)^\top w + \nabla_v\sigma(\rho r\sqrt{d})^\top w,$$

Likewise, we have

$$\frac{w^\top \nabla_u f}{(1-p)\lambda} \simeq p \cdot (\sigma(x_{\ell_*}^\top v) - \sigma(\rho r \sqrt{d})) x_{\ell_*}^\top w + \sigma(\rho r \sqrt{d}) r u^\top w - (p\sigma(x_{\ell_*}^\top v) + (1-p)\sigma(\rho r \sqrt{d})) r u^\top w$$

$$= p \cdot (\sigma(x_{\ell_*}^\top v) - \sigma(\rho r \sqrt{d})) x_{\ell_*}^\top w - p \cdot (\sigma(x_{\ell_*}^\top v) - \sigma(\rho r \sqrt{d})) r u^\top w,$$

$$\frac{w^\top \nabla_u f}{p(1-p)\lambda} \simeq (\sigma(x_{\ell_*}^\top v) - \sigma(\rho r \sqrt{d})) x_{\ell_*}^\top w - (\sigma(x_{\ell_*}^\top v) - \sigma(\rho r \sqrt{d})) r u^\top w$$

$$= \sigma_{\gamma,\zeta,\rho}(z^\top v)(c u_*^\top w - r u^\top w)\sqrt{d} + \gamma \sigma_{\gamma,\zeta,\rho}(z^\top v)(z^\top w).$$

Putting things together, we have shown that

$$\nabla_v f^\top w \simeq p\nabla_v(\sigma(x_{\ell_*}^\top v) - \sigma(\rho r \sqrt{d}))^\top w + \nabla_v \sigma(\rho r \sqrt{d})^\top w, \tag{80}$$

$$\frac{\nabla_u f^\top w}{p(1-p)\lambda} \simeq \sigma_{\gamma,\zeta,\rho}(z^\top v)(c u_*^\top w - r u^\top w)\sqrt{d} + \gamma \sigma_{\gamma,\zeta,\rho}(z^\top v)(z^\top w), \tag{81}$$

$$\delta \simeq p(\sigma(x_{\ell_*}^\top v) - \sigma(\rho r \sqrt{d})) - (\sigma(x_{\ell_*}^\top v_*) - \sigma(\rho r \sqrt{d})). \tag{82}$$

**Gradient w.r.t $u$ Parameter.** Proceeding from (81) and (82), we compute

$$\frac{w^\top \nabla_u R(u,v)}{2p(1-p)\lambda} = \mathbb{E}\left[\delta \cdot \frac{w^\top \nabla_u f}{p(1-p)\lambda}\right]$$

$$\simeq (c u_*^\top w - r u^\top w)\sqrt{d}\mathbb{E}[(p\sigma_{\gamma,\zeta,\rho}(z^\top v) - \sigma_{\gamma,0,\rho}(z^\top v_*))\sigma_{\gamma,\zeta,\rho}(z^\top v)]$$

$$+ \gamma \mathbb{E}[(p\sigma_{\gamma,\zeta,\rho}(z^\top v) - \sigma_{\gamma,0,\rho}(z^\top v_*))\sigma_{\gamma,\zeta,\rho}(z^\top v)(z^\top w)]$$

$$= (c u_*^\top w - r u^\top w)\sqrt{d}(p H_1(1,\zeta,\rho) - H_2(\nu,\zeta,\rho))$$

$$+ \gamma^2 \sqrt{d}(w^\top v_* + (1-2p)w^\top v)\partial_\rho[p H_1(1,\zeta,\rho) - H_2(\nu,\zeta,\rho)].$$

Since $r = 0$ by assumption, the Remark 3 tells us that all the $H_k$ functiosn no longer depend on $\rho$, and we get

$$\frac{1}{p(1-p)\lambda}\nabla_u R(u,v)^\top w \simeq B(\mu,\nu,\zeta)c\sqrt{d}u_*^\top w, \tag{83}$$

where $B(\mu,\nu,\zeta) := 2(p(\mu)h_1(1,\zeta) - h_2(\nu,\zeta))$ as usual. Recalling the definition

$$A(\mu) := p(\mu)(1-p(\mu))c\beta d = p(\mu)(1-p(\mu))c\lambda\sqrt{d},$$

we can write the above as

$$\nabla_u R(u,v)^\top w \simeq (w^\top u_*)A(\mu)B(\mu,\nu,\zeta) = T_1(\mu,\nu,\zeta)u_*^\top w. \tag{84}$$

This proves that $\nabla_u R(u,v) \simeq T_1(\mu,\nu,\zeta)u_*$ as claimed.

**Gradient w.r.t $v$ Parameter.** Using (80) and (82), one computes

$$\frac{w^\top \nabla_v R(u,v)}{2} = \mathbb{E}[\delta \cdot \nabla_v f]$$

$$\simeq \mathbb{E}[(p\sigma_{\gamma,\zeta,\rho}(z^\top v) - \sigma_{\gamma,0,\rho}(z^\top v_*))(p\nabla_v \sigma_{\gamma,\zeta,\rho}(z^\top v)^\top w + \nabla_v \sigma(\rho r \sqrt{d})^\top w)]$$

$$= p^2 w^\top \mathbb{E}[\sigma_{\gamma,\zeta,\rho}(z^\top v)\nabla_v \sigma_{\gamma,\zeta,\rho}(z^\top v)] + p\mathbb{E}[\sigma_{\gamma,\zeta,\rho}(z^\top v)]\nabla_v \sigma(\rho r \sqrt{d})^\top w$$

$$- pw^\top \mathbb{E}[\sigma_{\gamma,0,\rho}(z^\top v_*)\nabla_v \sigma_{\gamma,\zeta,\rho}(z^\top v)] - \mathbb{E}[\sigma_{\gamma,0,\rho}(z^\top v_*)]\nabla_v \sigma(\rho r \sqrt{d})^\top w$$

Now, using the well know formulae $g\nabla g = (1/2)\nabla g^2$ and $g\nabla h = \nabla gh - h\nabla g$, we have

$$2\mathbb{E}[\sigma_{\gamma,\zeta,\rho}(z^\top v)\nabla_v \sigma_{\gamma,\zeta,\rho}(z^\top v)] = \nabla_v \mathbb{E}[\sigma_{\gamma,\zeta,\rho}(z^\top v)^2] = \nabla_v H_1(1,\zeta,\rho)$$

$$= \partial_\zeta H_1(1,\zeta,\rho)u_* + \partial_\rho H(1,\zeta,\rho)u,$$

$$\mathbb{E}[\sigma_{\gamma,0,\rho}(z^\top v_*)\nabla_v \sigma_{\gamma,\zeta,\rho}(z^\top v)] = \nabla_v H_2(\nu,\zeta,\rho) - \mathbb{E}[\sigma_{\gamma,\zeta,\rho}(z^\top v)\nabla_v \sigma_{\gamma,0,\rho}(z^\top v_*)]$$

$$= \partial_\nu H_2(\nu,\zeta,\rho)v_* + \partial_\zeta H_2(\nu,\zeta,\rho)u_* + \partial_\rho H_2(\nu,\zeta,\rho)u$$

$$- \mathbb{E}[\sigma_{\gamma,\zeta,\rho}(z^\top v)\nabla_v \sigma_{\gamma,0,\rho}(z^\top v_*)].$$

Observe that $\nabla_v \sigma_{\gamma,0,\rho}(z^\top v_*) = -\nabla_v \sigma(\rho r \sqrt{d})$, and so

$$\mathbb{E}[\sigma_{\gamma,\zeta,\rho}(z^\top v)\nabla_v \sigma_{\gamma,0,\rho}(z^\top v_*)] = \mathbb{E}[\sigma_{\gamma,\zeta,\rho}(z^\top v)]\nabla_v \sigma(\rho r \sqrt{d}). \tag{85}$$

Putting everything (once again under the assumption that $r = 0$), we get

$$\begin{aligned}
\nabla_v R(u,v)^\top w &\simeq p\nabla_v(ph_1(1,\zeta) - 2h_2(\nu,\zeta))^\top w \\
&= p[(\partial_\zeta h_1(1,\zeta) - 2\partial_\zeta h_2(\nu,\zeta,\rho))u_* - 2\partial_\nu h_2(\nu,\zeta)v_*]^\top w \\
&= (T_3(\mu,\nu,\zeta)u_* + T_2(\mu,\nu,\zeta)v_*)^\top w.
\end{aligned}$$

We conclude that $\nabla_v R(u,v) \simeq T_3(\mu,\nu,\zeta)u_* + T_2(\mu,\nu,\zeta)v_*$ as claimed. $\qquad\square$

### E.3 Proof of Corollary 2

From (30), the dynamics for $(\mu,\nu,\eta,\zeta$ is given by

$$\begin{aligned}
\dot{\mu}(t) &= u_*^\top \dot{u}(t) = -u_* P_{u(t)}^\top \nabla_u R(u(t),v(t)), \\
\dot{\nu}(t) &= v_*^\top \dot{v}(t) = -u_* P_{v(t)}^\top \nabla_v R(u(t),v(t)), \\
\dot{\eta}(t) &= v_*^\top \dot{u}(t) = -v_* P_{u(t)}^\top \nabla_u R(u(t),v(t)), \\
\dot{\zeta}(t) &= u_*^\top \dot{v}(t) = -u_* P_{v(t)}^\top \nabla_v R(u(t),v(t)).
\end{aligned}$$

On the other hand, one computes

$$\begin{aligned}
P_u^\perp \nabla_u R &= T_1 \cdot (u_* - \mu u), \quad P_v^\perp \nabla_v R = T_3 \cdot (u_* - \zeta v) + T_2 \cdot (v_* - \nu v), \\
u_*^\top P_u^\perp \nabla_u R &= T_1 \cdot (1 - \mu^2) := -F_1, \quad u_*^\top P_v^\perp \nabla_v R = T_3 \cdot (1 - \zeta^2) - T_2\zeta\nu =: -F_4, \\
v_*^\top P_u^\perp \nabla_u R &= -T_1\eta\mu =: -F_3, \quad v_*^\top P_v^\perp \nabla_v R = -T_3\zeta\nu + T_2 \cdot (1 - \nu^2) =: -F_2,
\end{aligned}$$

and the result follows. $\qquad\square$

## F  Proofs of Classification Theorems

For ease of notation, we shall use the following shorthand

$$A(\mu) := p(\mu)(1 - p(\mu))c\beta d, \quad B(\mu,\nu) := B(\mu,\nu,0), \quad T_k(\mu,\nu) := T_k(\mu,\nu,0), \tag{86}$$

$$F_k(\mu,\nu) := F_k(\mu,\nu,0), \text{ for } k = 1,2,4; \quad F_3(\mu,\nu) := F_3(\mu,\nu,0,0). \tag{87}$$

### F.1  Jacobian Matrices

Classical theory of dynamical systems tells us that the classification of the different stationary points $(\mu,\nu) \in E$ can be done by studying the signs of the real parts of the eigenvalues of the Jacobian matrices

$$J(\mu,\nu) := \begin{bmatrix} \partial_\mu T_1(\mu,\nu) & \partial_\mu T_2(\mu,\nu) \\ \partial_\nu T_1(\mu,\nu) & \partial_\nu T_2(\mu,\nu) \end{bmatrix}. \tag{88}$$

One can use these Jacobian matrices to classify the stationary points as sources ($J(\mu,\nu)$ only has positive eigenvalues), sinks ($J(\mu,\nu)$ only has negative eigenvalues), and saddles ($J(\mu,\nu)$ has a positive and a negative eigenvalue).

Let us now compute the entries of each $J(\mu,\nu)$. From the definition of $T_1$ and $T_2$, we have

$$\partial_\mu T_1(\mu,\nu) = 2(c\beta d)^2 \left( (2p-1)p(1-p)B(\mu,\nu)/2 + p(1-p)p'\bar{\sigma}_\gamma(1) \right) \tag{89}$$

$$= 2(c\beta d)^2 p(1-p) \left( (2p-1)B(\mu,\nu)/2 + p(1-p)\bar{\sigma}_\gamma(1) \right), \tag{90}$$

$$\partial_\nu T_1(\mu,\nu) = -2p(1-p)c\beta d \cdot \bar{\sigma}'_\gamma(\nu), \tag{91}$$

$$\partial_\mu T_2(\mu,\nu) = -2p(1-p)c\beta d \cdot \bar{\sigma}'_\gamma(\nu), \tag{92}$$

$$\partial_\nu T_2(\mu,\nu) = -2p\bar{\sigma}''_\gamma(\nu). \tag{93}$$

Thus, for any stationary point $(\mu, \nu) \in E$, we have

$$\partial_\mu F_1(\mu, \nu) = -(1 - \mu^2)\partial_\mu T_1(\mu, \nu) + 2T_1(\mu, \nu)\mu$$
$$= \begin{cases} 2T_1(\mu, \nu)\mu = 2A(\mu)B(\mu, \nu)\mu, & \text{if } (\mu, \nu) \in E_1 \cup E_2, \\ -(1 - \mu^2)\partial_\mu T_1(\mu, \nu) = -2(1 - \mu^2)A(\mu)^2\bar\sigma_\gamma(1) < 0, & \text{if } (\mu, \nu) \in E_3 \cup E_4, \end{cases}$$
$$\partial_\nu F_1(\mu, \nu) = -(1 - \nu^2)\partial_\nu T_1(\mu, \nu) = 0,$$
$$\partial_\mu F_2(\mu, \nu) = -(1 - \mu^2)\partial_\mu T_2(\mu, \nu) = 0,$$
$$\partial_\nu F_2(\mu, \nu) = -(1 - \nu^2)\partial_\nu T_2(\mu, \nu) + 2T_2(\mu, \nu)\nu,$$
$$= \begin{cases} 2T_2(\mu, \nu)\nu = -4p(\mu)\bar\sigma_\gamma'(\nu)\nu, & \text{if } (\mu, \nu) \in E_1 \cup E_3, \\ -(1 - \nu^2)\partial_\nu T_2(\mu, \nu) = 2(1 - \nu^2)p(\mu)\bar\sigma_\gamma''(\nu), & \text{if } (\mu, \nu) \in E_2 \cup E_4. \end{cases}$$

We deduce the following Lemma which will be crucial in the proofs.

**Lemma 1.** *Recall the definition of $A(\mu)$ and $B(\mu, \nu)$ from* (86)*. For any stationary point $(\mu, \nu) \in E = E_1 \cup E_2 \cup E_3 \cup E_4$ of the dynamics* (35)*, the Jacobian matrix* (88) *has the following form.*

- *If $(\mu, \nu) \in E_1$, then $J(\mu, \nu) = \begin{bmatrix} 2A(\mu)B(\mu, \nu)\mu & 0 \\ 0 & -4p(\mu)\bar\sigma_\gamma'(\nu)\nu \end{bmatrix}$.*

- *If $(\mu, \nu) \in E_2$, then $J(\mu, \nu) = \begin{bmatrix} 2A(\mu)B(\mu, \nu)\mu & 0 \\ 0 & 2(1 - \nu^2)p(\mu)\bar\sigma_\gamma''(\nu) \end{bmatrix}$.*

- *If $(\mu, \nu) \in E_3$, then $J(\mu, \nu) = \begin{bmatrix} -2(1 - \mu^2)A(\mu)^2\bar\sigma_\gamma(1) & 0 \\ 0 & -4p(\mu)\bar\sigma_\gamma'(\nu)\nu \end{bmatrix}$.*

- *If $(\mu, \nu) \in E_4$, then $J(\mu, \nu) = \begin{bmatrix} -2(1 - \mu^2)A(\mu)^2\bar\sigma_\gamma(1) & 0 \\ 0 & 2(1 - \nu^2)p(\mu)\bar\sigma_\gamma''(\nu) \end{bmatrix}$.*

### F.2  Proof of Proposition 9

Thanks to Lemma 1, the Jcobian matrix is given by

$$J(1, 1) = \begin{bmatrix} 2A(1)B(1, 1) & 0 \\ 0 & -4p(1)\bar\sigma_\gamma'(1) \end{bmatrix}.$$

Note that by definition the $A$ and $B$ functions in from (86) that

$$A(1)B(1, 1) = p(1)(1 - p(1))c\beta d \cdot 2(p(1) - 1)(= \bar\sigma_\gamma(1) < 0$$

because $p(1) \in (0, 1)$ and $\bar\sigma_\gamma(1) > 0$. On the other hand, $-4p(1)\bar\sigma_\gamma'(1)$ is negative since $\bar\sigma_\gamma'(1) = \sum_{n \geq 1} nc_n^2 > 0$. We deduce that $J(1, 1)$ has only negative eigenvalues, and the result follows. $\square$

### F.3  Proof of Proposition 10

We now consider a few important choices for the link function $\sigma$, and provide a complete classification of all the stationary points of the induced $(\mu, \nu)$-dynamics. The picture will of course depend on the underlying link function $\sigma$.

**Linear Link Function.**  Consider the case $\sigma(t) \equiv t$. It is clear that

$$\bar\sigma_\gamma(\nu) \equiv \gamma\nu, \quad \bar\sigma_\gamma'(\nu) \equiv \gamma, \quad \bar\sigma_\gamma''(\nu) \equiv 0.$$

It is then easy to see that $E_2 = E_3 = E_4 = \emptyset$, and so the set of stationary points is

$$E = \cup_k E_k = E_1 = \{(-1, -1), (-1, 1), (1, -1), (1, 1)\}.$$

Observe that for any stationary point $(\mu, \nu) \in E$, one has

$$\partial_\mu F_1(\mu, \nu) = 2T_1(\mu, \nu)\mu, \quad \partial_\nu F_2(\mu, \nu) = 2T_2(\mu, \nu)\nu, \quad \partial_\nu F_1(\mu, \nu) = \partial_\mu F_2(\mu, \nu) = 0.$$

Now, $T_1(\pm 1, \nu) = (p(\pm 1) - \nu)A$, $T_2(\pm 1, \nu) = -2\gamma p(\pm 1)$, with $A := p(1)(1 - p(1))c\beta d > 0$. The Jacobian matrices at each stationary point is then given by

$$J(-1, -1) = \begin{bmatrix} -2(p(-1) + 1)A\gamma & 0 \\ 0 & 2\gamma^2 p(-1) \end{bmatrix}, \tag{94}$$

$$J(-1, 1) = \begin{bmatrix} -2(p(-1) - 1)A\gamma & 0 \\ 0 & -2\gamma^2 p(-1) \end{bmatrix}, \tag{95}$$

$$J(1, -1) = \begin{bmatrix} 2(p(1) + 1)A\gamma & 0 \\ 0 & 2\gamma^2 p(1) \end{bmatrix}, \tag{96}$$

$$J(1, 1) = \begin{bmatrix} 2(p(1) - 1)A\gamma & 0 \\ 0 & -2\gamma^2 p(1) \end{bmatrix}, \tag{97}$$

$$\tag{98}$$

Since $A > 0$, $p(\mu) + 1 > 0$ and $p(\mu) - 1 < 0$ for all $\mu$, we deduce that

- $J(-1, \pm 1)$ each have one negative and positive eigenvalue: these are saddles.
- $J(1, -1)$ has both eigenvalues positive: this is a source.
- $J(1, 1)$ has both eigenvalues negative: this is a sink.

This proves Proposition 10(A). $\qquad\square$

**Quadratic Link Function.** Here $\sigma(t) \equiv t^2$ and so thanks to Proposition 11, $\bar{\sigma}_\gamma(\nu) = \gamma^4(1 + 2\nu^2)$. Consequently, we have

$$\bar{\sigma}_\gamma(\pm 1) = 2\gamma^4 > 0, \quad \bar{\sigma}'_\gamma(\nu) \equiv 4\gamma^4\nu, \quad \bar{\sigma}''_\gamma(\nu) \equiv 4\gamma^4. \tag{99}$$

It is then clear that $E_2 := \{(\pm 1, \nu) \mid \nu \in (-1, 1), \bar{\sigma}'(\nu) = 0\} = \{(\pm 1, 0)\}$.

We now consider $E_3 := \{(\mu, \pm 1) \mid p(\mu)(\bar{\sigma}'_\gamma(\nu) - \bar{\sigma}_\gamma(\pm 1) = 0\}$. Now, $0 = p(\mu)\bar{\sigma}_\gamma(1) - \bar{\sigma}_\gamma(\pm 1)$ iff

$$p(\mu) = \frac{\bar{\sigma}_\gamma(\pm 1)}{\bar{\sigma}_\gamma(1)} = 1,$$

which is impossible. Thus, $E_3 = \emptyset$.

Finally, we compute $E_4$. By definition,

$$E_4 = \{(\mu, \nu) \in (-1, 1)^2 \mid \bar{\sigma}'_\gamma(\nu) = p(\mu)\bar{\sigma}'_\gamma(1) - \bar{\sigma}_\gamma(\nu)\} = \{(\mu, 0) \mid p(\mu)\bar{\sigma}'_\gamma(1) - \bar{\sigma}_\gamma(0) = 0\}.$$

Now, $0 = p(\mu)\bar{\sigma}_\gamma(1) - \bar{\sigma}_\gamma(0)$ iff

$$p(\mu) = \frac{\bar{\sigma}_\gamma(0)}{\bar{\sigma}_\gamma(1)} = \frac{1}{2}, \text{ i.e., } \mu = \psi.$$

which is feasible iff $\psi < 1$.

Therefore, the stationary points are $E = E_1 \cup E_2 \cup E_4$, with $E_1 = \{(\pm 1, \pm 1)\}$, $E_2 = \{(\pm 1, 0)\}$, and $E_4 = \{(\psi, 0)\}$. Recall $A(\mu)$ and $B(\mu, \nu)$ from (86). We have the following classification

- $(\pm 1, \pm 1)$ at each such stationary point, the Jacobian is given by

$$J(\mu, \nu) = \begin{bmatrix} 2A(\mu)B(\mu, \nu)\mu & 0 \\ 0 & -4p(\mu)\bar{\sigma}'_\gamma(\nu)\nu \end{bmatrix} = \begin{bmatrix} 2A(\mu)B(\mu, \nu)\mu & 0 \\ 0 & -16p(\mu)\gamma^4 \end{bmatrix}$$

Note, that for any $\mu \in [-1, 1]$, it holds that

$$B(\mu, \pm 1) = 2(p(\mu)\bar{\sigma}_\gamma(1) - \bar{\sigma}_\gamma(\pm 1)) = 2(p(\mu) - 1)\bar{\sigma}_\gamma(1) < 0,$$

since $\bar{\sigma}_\gamma(-1) = \bar{\sigma}_\gamma(1) > 0$. Thus, $\text{sign}(A(\mu)B(\mu, \nu)\mu) = -\text{sign}(\mu)$ for all $(\mu, \nu) \in \{(\pm 1, \pm 1)\}$. Thus, $J(-1, \pm 1)$ each have one negative and positive eigenvalues and $J(1, \pm 1)$ each have only positive eigenvalues. We conclude that $(-1, \pm 1)$ are saddles while, $(1, \pm 1)$ are sinks (stable stationary points). It should comfort the reader to know that $(1, \pm 1)$ are equivalent representations of the oracle (i.e. Bayes-optimal) parameters $(u_*, v_*)$ because due to the evenness of the quadratic link function, replacing $v_*$ by $-v_*$ doesn't change the oracle model.

- $(\pm 1, 0)$: Here, $J(\mu, \nu) = \begin{bmatrix} 2A(\mu)B(\mu,0)\mu & 0 \\ 0 & 2p(\mu)\bar\sigma_\gamma''(0) \end{bmatrix}$. In particular, we see that $J(-1,0)$ has one negative and one positive eigenvalue, while $(1,0)$ has only positive eigenvalues. We conclude that $(-1,0)$ is a saddle, while $(1,0)$ is a source.

- $(\psi, 0)$: This is a stationary point only if $0 \le \psi < 1$. In that case, we have

$$J(\psi,0) = \begin{bmatrix} -2(1-\psi^2)A(\psi)^2\bar\sigma_\gamma(1) & 0 \\ 0 & \bar\sigma_\gamma''(0) \end{bmatrix} = \begin{bmatrix} -2(1-\psi^2)A(\psi)^2\bar\sigma_\gamma(1) & 0 \\ 0 & 4\gamma^4 \end{bmatrix}.$$

Thus, $J(\psi,0)$ has one negative and positive eigenvalue. We conclude that $(\psi,0)$ is a saddle.

This proves Proposition 10(B). $\qquad\square$

**ReLU Link Function.** Consider the case where $\sigma(t) \equiv (t)_+$, thanks to Proposition 11, we have

$$\bar\sigma_\gamma(\nu) := \frac{\gamma^2}{2\pi}(\sqrt{1-\nu^2} + \nu\arccos(-\nu))$$

One then readily computes

$$\bar\sigma_\gamma(1) = \frac{\gamma^2}{2}, \quad \bar\sigma_\gamma(-1) = 0, \quad \bar\sigma_\gamma'(\nu) = \frac{\gamma^2}{2\pi}\arccos(-\nu),$$

and so

$$E_2 = \{(\pm 1, \nu) \mid \nu \in (-1,1),\ \bar\sigma_\gamma'(\nu) = 0\} = \{(\pm 1, \nu) \mid \nu \in (-1,1),\ \arccos(-\nu) = 0\} = \emptyset,$$

$$E_4 := \{(\mu, \nu) \in (-1,1)^2,\ p(\mu)\bar\sigma_\gamma(1) - \bar\sigma_\gamma(-1) = \bar\sigma_\gamma'(\nu) = 0\}$$

$$= \{(\mu, \nu) \in (-1,1)^2,\ p(\mu)\bar\sigma_\gamma(1) - \bar\sigma_\gamma(-1) = \arccos(-\nu) = 0\} = \emptyset,$$

$$E_3 := \{(\mu, -1) \mid \mu \in (-1,1),\ p(\mu)\bar\sigma_\gamma(1) - \bar\sigma_\gamma(-1) = 0\}$$

$$= \{(\mu, -1) \mid \mu \in (-1,1),\ p(\mu)\gamma^2/2 = 0\} = \emptyset.$$

Thus, the stationary points are $E = E_1 = \{(\pm 1, \pm 1)\}$. Now, for any $(\mu, \nu) \in E_1$, the Jacobian of the dynamics is

$$J(\mu, \nu) = \begin{bmatrix} 2A(\mu)B(\mu,\nu)\mu & 0 \\ 0 & -4p(\mu)\bar\sigma_\gamma'(\nu)\nu \end{bmatrix} = \begin{bmatrix} 2A(\mu)B(\mu,\nu)\mu & 0 \\ 0 & -2p(\mu)\gamma^2\nu\arccos(-\nu)/\pi \end{bmatrix}.$$

More explicitly, we have

$$J(\mu, -1) = \begin{bmatrix} 2A(\mu)B(\mu,\nu)\mu & 0 \\ 0 & 0 \end{bmatrix}, \quad J(\mu, 1) = \begin{bmatrix} 2A(\mu)B(\mu,\nu)\mu & 0 \\ 0 & -2p(\mu)\gamma^2 \end{bmatrix}. \qquad (100)$$

Since $B(\mu, 1) := 2(p(\mu)\bar\sigma_\gamma(1) - \bar\sigma_\gamma(1)) = -2(1-p(\mu))\bar\sigma_\gamma(1) < 0$ (because $\bar\sigma_\gamma(1) > 0$ under the ongoing constraints), we deduce that each of $J(\pm 1, -1)$ has one negative and one zero eigenvalue; while $J(-1, 1)$ has one positive and one negative eigenvalue; $J(1, 1)$ has only negative eigenvalues. We conclude $(1, 1)$ is a sink (stable stationary point), $(-1, 1)$ is a saddle.

This completes the proof of Proposition 10C. $\qquad\square$

## F.4 Proof of Proposition 4

For linear link function $\sigma(t) \equiv t$, we have for any $\mu, \nu, \zeta \in [-1, 1]$,

$$h_1(\nu, \zeta) = \gamma^2\nu + a^2, \quad h_2(\nu, \zeta) = \gamma^2\nu, \quad \text{with } a := c\zeta\sqrt{d},$$

$$\partial_\zeta h_1(1, \zeta) - 2\partial_\zeta h_2(\nu, \zeta) = 2a = 2c\zeta\sqrt{d}, \quad \partial_\nu h_2(\nu, \zeta) = \gamma^2.$$

We can thus simplify the auxiliary functions $F_1, F_2, F_3, F_4$ appearing in Section E.1 like so:

$$F_1(\mu, \nu, \zeta) = -(1-\mu^2)T_1, \qquad (101)$$

$$F_2(\mu, \nu, \zeta) = \zeta\nu T_3 - (1-\nu^2)T_2, \qquad (102)$$

$$F_3(\mu, \nu, \eta, \zeta) = -\eta\mu T_1, \qquad (103)$$

$$F_4(\mu, \nu, \zeta) = -(1-\zeta^2)T_3 + \zeta\nu T_2, \qquad (104)$$

$$T_2(\mu, \nu, \zeta) = -2p\gamma^2, \quad T_1(\mu, \nu, \zeta) = A(\mu)B(\mu, \nu, \zeta), \qquad (105)$$

$$B(\mu, \nu, \zeta) = 2(ph_1(1, \zeta) - h_2(\nu, \zeta)) = 2(p \cdot (\gamma^2 + a^2) - \gamma^2\nu), \qquad (106)$$

$$T_3(\mu, \nu, \zeta) = p(u)\partial_\zeta[h_1(1, \zeta) - 2h_2(\nu, \zeta)] = 2p(\mu)a = 2p(\mu)c\zeta\sqrt{d}. \qquad (107)$$

Now, the stationary points $(\mu, \nu, \eta, \zeta)$ of the equations of motion given in Proposition 2 are defined by the equations $F_1 = F_2 = F_3 = F_4 = 0$. We will prove that any such point must verify $\eta = \zeta = 0$.

We prove that $\zeta = 0$. Suppose on the contrary that $\zeta \neq 0$. Then, the equations $F_2 = F_4 = 0$ give $\nu = p\gamma^2/C$ and $\zeta^2 = 1 - (p\gamma^2)^2/C^2$, where $C := c\sqrt{d} \gg p\gamma^2$. This implies

$$B/2 = p \cdot (\gamma^2 + a^2) - \gamma^2 \nu = p \cdot (\gamma^2 + a^2 - \gamma^2/C) = p \cdot (\gamma^2(1 - 1/C) + a^2) > 0.$$

Thus, we must have $B \neq 0$, and so the equation $F_1 = 0$ gives $u^\top u_* = \mu = \pm 1$. Now, $\mu = u^\top u_* = \pm 1$ implies $u = \pm u_*$, and so $\zeta := v^\top u_* = \pm v^\top u = \pm \rho = 0$. By *reductio ad absurdum*, we must conclude $\zeta = 0$.

Note that with $\eta = 0$, the equation $F_2 = 0$ now gives $\nu = \pm 1$. This means $v = \pm v_*$ and so $\eta := u^\top v_* = \pm u^\top v = \pm \rho = 0$. We conclude that every stationary has $\eta = \zeta = 0$.

We now show that the only stationary points are $(\pm 1, \pm 1)$. Now, plugging $\eta = \zeta = 0$, the stationary point must satisfy

$$
\begin{aligned}
0 = F_1^0(\mu, \nu) &:= -(1 - \mu^2)T_1(\mu, \nu, 0) = -(1 - \mu^2)A(\mu)2(p(\mu)\bar{\sigma}_\gamma(1) - \bar{\sigma}_\gamma(\nu)) \\
&= -2A(\mu)(1 - \mu^2)(p(\mu) - \nu)\gamma^2, \\
0 = F_2^0(\mu, \nu) &:= -(1 - \nu^2)T_2(\mu, \nu, 0) = (1 - \nu^2)2p(\mu)\gamma^2.
\end{aligned}
$$

Because $p(\mu)\gamma^2 > 0$, second equation tells us that we must have $\nu = \pm 1$. Plugging this into the first equation and dividing through by the factor $2A(\mu) > 0$ gives $(1 - \mu^2)(p(\mu) \mp 1) = 0$. But $p(\mu) \mp 1 \neq 0$ because $p(\mu) \in (0, 1)$ for all $\mu$, and we conclude that $\mu = \pm 1$. This shows that the stationary points are $(\pm 1, \pm 1, 0, 0)$ as claimed.

The classification of the stationary points then follows from Proposition 10(A). $\qquad \square$

# G Proof of Proposition 8 (Convergence of PGD to Stationary Point)

## G.1 Step 1: A Descent Lemma

We shall need the following regularity assumption for the link function $\sigma$.

**Assumption 1.** *The link function $\sigma$ is $C^2$ on $\mathbb{R}$ with $\|\sigma'\|_\infty, \|\sigma''\|_\infty < \infty$. The case of ReLU activation needs special treatment (not provided here).*

One can show that on $S_{d-1}^2$, the $R$ functional

$$
\begin{aligned}
R(u, v) &:= p^2 H_1(1, \zeta, \rho) - 2p H_2(\nu, \zeta, \rho) + H_1(1, 0, \rho), \\
&\text{with } \mu := u^\top u_*, \quad \nu := v^\top v_*, \quad \zeta := v^\top u_*, \quad \rho := v^\top u, \\
&p := (1 + e^{(c\mu - \psi)\beta d})^{-1},
\end{aligned}
\tag{108}
$$

is $L$-smooth on $S_{d-1}^2$ for some finite $L > 0$. Now, consider the following canonical the extension $r$ of $R$ from $S_{d-1}^2$ to all of $\mathbb{R}^{2d}$

$$r(u, v) := R(u/\|u\|, v/\|v\|).$$

Note that $\nabla_u r(u, v) = P_u^\perp \nabla_u R(u, v)$ for all $(u, v) \in S_{d-1}^2 \subseteq S_{d-1}^2(\delta) \subseteq \mathbb{R}^{2d}$. Then, one can show that for any $\delta \in (0, 1]$, the functional $r$ is $L_\delta$-smooth with $L_\delta := 4L/(1 - \delta)^2$, on the tube

$$S_{d-1}^2(\delta) := \{(u, v) \in \mathbb{R}^{2d} \mid 1 - \delta \leq \|u\|, \|v\| \leq 1 + \delta\}.$$

This means that

$$r(u', v') \leq r(u, v) + \langle \nabla r(u, v), \Delta \rangle + \frac{L_\delta}{2}\|\Delta\|^2, \tag{109}$$

for any $u, v, u', v' \in S_{d-1}^2(\delta)$, with $\Delta = (u' - u, v' - v) \in \mathbb{R}^{2d}$. Furthermore, because $r$ is radially symmetric, we know that $\nabla_u r(u, v) \perp u$ and $\nabla_v r(u, v) \perp v$ for all non-zero $u, v \in \mathbb{R}^d$.

Now, define $g_k = (a_k, b_k)$, where $a_k, b_k \in \mathbb{R}^d$ are defined by

$$a_k := \nabla_u r(u_k, v_k) = P_{u_k}^\perp \nabla_u R(u, v), \quad b_k := \nabla_v r(u_k, v_k) = P_{v_k}^\perp \nabla_v R(u_k, v_k).$$

In (109) above, taking $u = u_k$, $v = v_k$, $u' = \tilde{u}_{k+1} := u_k - sa_k$ (as in (28)), and $v' := \tilde{v}_{k+1} = u_k - sg_k$, we get

$$r(\tilde{u}_{k+1}, \tilde{v}_{k+1}) \leq r(u_k, v_k) - s\langle \nabla r(u_k, v_k), g_k \rangle + \frac{s^2 L_\delta}{2}\|g_k\|^2$$

$$= r(u_k, v_k) - s\|g_k\|^2 + \frac{s^2 L_\delta}{2}\|g_k\|^2 \qquad (110)$$

$$= r(u_k, v_k) - (s - s^2 L_\delta/2)\|g_k\|^2.$$

We shall now control the deviation of $(u_{k+1}, v_{k+1})$ from $(\tilde{u}_{k+1}, \tilde{v}_{k+1})$. By definition, we have

$$\|u_{k+1} - \tilde{u}_{k+1}\| := \|(\frac{1}{\|\tilde{u}_{k+1}\|} - 1)\tilde{u}_{k+1}\| = |\|\tilde{u}_{k+1}\| - 1|$$

$$\|v_{k+1} - \tilde{v}_{k+1}\| := \|(\frac{1}{\|\tilde{v}_{k+1}\|} - 1)\tilde{v}_{k+1}\| = |\|\tilde{v}_{k+1}\| - 1|. \qquad (111)$$

Now, because $\tilde{u}_{k+1} = u_k - sg_{u_k}$ with $\|u_k\| = 1$ and $u_k \perp g_{u_k}$, we have

$$1 \leq \|\tilde{u}_{k+1}\| = \sqrt{1 + s^2\|g_{u_k}\|^2} \leq 1 + \frac{s^2}{2}\|g_{u_k}\|^2,$$

where the last step uses the elementary inequality $\sqrt{1+a} \leq 1 + a/2$ for all $a \geq 0$. Similarly, for $v_{k+1}$ we have

$$1 \leq \|\tilde{v}_{k+1}\| = \sqrt{1 + s^2\|g_{v_k}\|^2} \leq 1 + \frac{s^2}{2}\|g_{v_k}\|^2.$$

On the other hand, (28) and (29) give

$$u_{k+1} - \tilde{u}_{k+1} = (\frac{1}{\|\tilde{u}_{k+1}\|} - 1)\tilde{u}_{k+1}, \; v_{k+1} - \tilde{v}_{k+1} = (\frac{1}{\|\tilde{v}_{k+1}\|} - 1)\tilde{v}_{k+1}$$

We deduce that

$$\|u_{k+1} - \tilde{u}_{k+1}\| = |\|\tilde{u}_{k+1}\| - 1| \leq \frac{s^2}{2}\|a_k\|^2,$$

$$\|v_{k+1} - \tilde{v}_{k+1}\| = |\|\tilde{v}_{k+1}\| - 1| \leq \frac{s^2}{2}\|b_k\|^2. \qquad (112)$$

Using this in (109) above gives with $u = \tilde{u}_{k+1}$, $v = \tilde{v}_{k+1}$, $u' := u_{k+1} = \tilde{u}_{k+1}$, $v' := u_{k+1}$, so that $\Delta = \zeta_k := (u_{k+1} - \tilde{u}_{k+1}, v_{k+1} - \tilde{v}_{k+1})$ gives

$$r(u_{k+1}, v_{k+1}) \leq r(\tilde{u}_{k+1}, \tilde{v}_{k+1}) + \langle r(\tilde{u}_{k+1}, \tilde{v}_{k+1}), \zeta_k \rangle + \frac{L_\delta}{2}\|\zeta_k\|^2$$

$$= r(\tilde{u}_{k+1}, \tilde{v}_{k+1}) + \frac{s^4 L_\delta}{8}\|g_k\|, \qquad (113)$$

where we have used the fact that $\nabla r(\tilde{u}_{k+1}, \tilde{v}_{k+1}) \perp \zeta_k$. Combining with (110) above gives

$$r(u_{k+1}, v_{k+1}) \leq r(u_k, v_k) - (s - s^2 L_\delta/2)\|g_k\|^2 + \frac{s^4 L_\delta}{8}\|g_k\|^4.$$

Now, one can show that $\|\nabla r(u, v)\|_\infty \leq M_\delta < \infty$ uniformly on $S_{d-1}(\delta)$. Thus, if $0 < s < 1/M_\delta$, then $s^4\|g_k\|^4 = (s^2\|g_k\|^2)s^2\|g_k\|^2 \leq s^2\|g_k\|^2$, and we get

$$r(u_{k+1}, v_{k+1}) \leq r(u_k, v_k) - (s - s^2 L_\delta/2 - s^2 L_\delta/8)\|g_k\|^2$$

$$= r(u_k, v_k) - (s - 5s^2 L_\delta/8)\|g_k\|^2 \qquad (114)$$

$$\leq R(u_k, v_k) - \frac{11}{16}sL_\delta\|g_k\|^2 \leq r(u_k, v_k) - \frac{1}{4}sL_\delta\|g_k\|^2,$$

provided the stepsize $s$ is sufficiently small in the sense that

$$0 < s < \min(1/(2L_\delta), 1/M_\delta).$$

Noting that $r = R$ on $S_{d-1}^2 \subseteq S_{d-1}^2(\delta)$, we get the following descent lemma.

**Lemma 2.** *If the step size $s$ is sufficiently small in the sense that $0 < s < \min(1/(2L_\delta), M_\delta)$, then*

$$R_{k+1} \leq R_k - \frac{sL_\delta}{4}\|g_k\|^2, \text{ with } R_k := R(u_k, v_k), \qquad (115)$$

*where we recall that $g_k = (a_k, b_k)$, $a_k = P_{u_k}^\perp \nabla_u R(u_k, v_k)$, $b_k = P_{v_k}^\perp \nabla_v R(u_k, v_k)$.*

## G.2 Step 2: Convergence to Stationary Point

The above inequality can be rewritten as $sL_\delta\|g_k\|^2/4 \le R_k - R_{k+1}$, and summing both sides gives

$$\frac{sL_\delta}{4}\sum_{k=0}^{K-1}\|g_k\|^2 \le R_0 - R_K \le R_0 - R_{min} < \infty,$$

where $R_{min} := \min_{u,v \in S_{d-1}} R(u,v) = 0$. We deduce that $\|g_k\| \to 0$, and so $g_k \to 0$ in the limit $k \to 0$. Now, one computes

$$\begin{aligned}
\|u_{k+1} - u_k\| &= \|\frac{\tilde{u}_{k+1}}{\|\tilde{u}_{k+1}\|} - u_k\| \\
&= \|\frac{u_k - sa_k}{\|\tilde{u}_{k+1}\|} - u_k\| = \|\frac{1 - \|\tilde{u}_{k+1}\|}{\|\tilde{u}_{k+1}\|}u_k - \frac{s}{\|\tilde{u}_{k+1}\|}a_k\| \\
&\le (|1 - \|\tilde{u}_{k+1}\|| + s\|a_k\|)\frac{1}{\|\tilde{u}_{k+1}\|} \\
&\le \frac{s^2}{2}\|a_k\|^2 + s\|a_k\| \text{ because } \|\tilde{u}_{k+1}\| \ge 1 \\
&\le \frac{s}{2}\|a_k\| + s\|a_k\| \text{ because } s\|a_k\| \le 1 \\
&= \frac{3}{2}s\|a_k\|.
\end{aligned}$$

Analogously, we get $\|v_{k+1} - v_k\| \le (3/2)s\|b_k\|$. Combining gives

$$\|u_{k+1} - u_k\|^2 + \|v_{k+1} - v_k\|^2 \le \frac{9}{4}s^2\|g_k\|^2 \to 0.$$

We deduce that the PGD iterates $(u_k, v_k)$ given in (29) form a Cauchy sequence in $S_{d-1}^2$. Due to completeness of $S_{d-1}^2$, this sequence has a limit $(u_\infty, v_\infty) \in S_{d-1}^2$. We now show that $(u_\infty, v_\infty)$ is a stationary point of the risk functional $R$.

For simplicity of presentation, we focus on the case of linear link function $\sigma(t) \equiv t$.

Thanks to Proposition 3, if $(u_0, v_0) \in \mathcal{M}$, then $(u_k, v_k) \in \mathcal{M}$ for all $k$ and one computes,

$$\begin{aligned}
a_k &= (I - u_k u_k^\top)\nabla_u R(u_k, v_k) = (I - u_k u_k^\top)T_1 u_* = (u_* - \mu_k u_k)T_1, \\
b_k &= (I - v_k v_k^\top)\nabla_v R(u_k, v_k) = (I - v_k v_k^\top)T_2 v_* = (v_* - \nu_k v_k)T_2,
\end{aligned}\tag{116}$$

where $T_j = T_j(\mu_k, \nu_k, 0)$ for $j = 1, 2, 3$, as defined in Appendix E.1, with $\mu_k := u_k^\top u_*$ and $\nu_k := v_k^\top v_*$. Note that we have used the fact that $T_3(\mu_k, \nu_k, 0) = 0$ thanks to the equation (107). We deduce that

$$\|g_k\|^2 = \|a_k\|^2 + \|b_k\|^2 = (1 - \mu_k^2)T_1^2 + (1 - \nu_k^2)T_2^2.$$

Thus, the limit point $(u_\infty, v_\infty)$ is such that

$$(1 - \mu_\infty^2)T_1(\mu_\infty, \nu_\infty, 0) = 0, \quad (1 - \nu_\infty^2)T_2(\mu_\infty, \nu_\infty, 0) = 0.$$

This is precisely the characterization of stationary points risk functional $R$ established in Proposition 7. This concludes the proof of Proposition 8. $\qquad\square$

