# OpenReview forum: "Understanding Softmax Attention Layers:\\ Exact Mean-Field Analysis on a Toy Problem"
_NeurIPS.cc/2025/Conference — NeurIPS 2025 poster_

### Official Review · Reviewer_mLDi · 2025-06-26

**Clarity:** 4
**Significance:** 4
**Originality:** 3
**Rating:** 5
**Confidence:** 4

**Summary:**

This paper develops an exact mean-field theory for a single-layer softmax self-attention network on the *single-location regression task*. Building on ideas from the random-energy model, they derive a closed-form expression for the population risk in the high-dimensional limit where the sequence length grows as $\log L = \mathcal{O}(d)$. They uncover a sharp phase transition that delineates learnable and non-learnable regions as a function of the feature-noise level $\gamma$ and the length parameter $\alpha$. Finally, for linear link functions they prove that Riemannian gradient descent converges from arbitrary initialization to the teacher parameters, which are shown to be the unique stable fixed point of the dynamics.

**Questions:**

- **Task correspondence.** Which concrete NLP tasks (e.g., keyword spotting, pointer networks, span extraction) map most closely to the single-location regression problem?
- **Finite-sample effects.** The optimisation study uses population risk. How do you expect finite-data noise or mini-batch updates to alter the dynamics? Could replica or dynamical mean-field tools quantify these effects?
- **Sphere constraint.** How crucial is the unit-sphere constraint on $u$ and $v$ ?  Could the analysis extend to unconstrained parameters with $l^{2}$ regularisation, and what obstacles would arise?

**Ethical Concerns:**

["NO or VERY MINOR ethics concerns only"]

**Final Justification:**

My questions have been resolved through the rebuttal. I will maintain my current rating of “accept.”

**Limitations:**

yes

**Paper Formatting Concerns:**

No major formatting issues.

**Quality:**

4

**Strengths And Weaknesses:**

## Strengths

- **Realistic modelling of attention**. Deriving the population risk for a softmax layer, rather than a decorrelated surrogate in previous study, makes the analysis markedly more linked to real Transformers.
- **Scales to long contexts**. The regime $\log L = \mathcal{O}(d)$ captures the much longer sequences found in modern NLP workloads, which is more realistic than earlier work restricted to $L=o(d)$.
- **Explicit learnability boundary**: The closed-form condition that separates solvable from unsolvable regions offers valuable theoretical guidance.
- **Strong optimisation guarantee**: For $\sigma(t)=t$, the proof that the teacher parameters form the sole attractor, without any special initialization, significantly strengthens prior results.

## Weakness

- **independence and Gaussianity assumption**. Real token embeddings are correlated and far from Gaussian. Although the paper moves the theory closer to practice, lifting this assumption remains an important next step.
- **Dynamics only under Condition 3.** Convergence is analysed in the frozen regime ($r=0$). A more detailed discussion of why dropping Condition 3 is technically hard—and what partial results might still be feasible—would be helpful.

---

> ### Author Rebuttal · Authors · 2025-07-31
>
> ## Response to Reviewer mLDi
> We thank the reviewer for their insightful feedback and questions.
>
> >***Task correspondence.** Which concrete NLP tasks (e.g., keyword spotting, pointer networks, span extraction) map most closely to the single-location regression problem?*
>
> All the examples mentioned by the reviewer are captured by the theoretical setup considered in our work. Let us also mention: in-context learning, topic understanding, sentiment analysis, etc. as they all have some kind of , token-wise sparsity and internal linear representation.
>
> >***Finite-sample effects.** The optimisation study uses population risk. How do you expect finite-data noise or mini-batch updates to alter the dynamics? Could replica or dynamical mean-field tools quantify these effects?*
>
> The reviewer's comment is spot on. Indeed, as mentioned in the **Concluding Remarks**, a future extension of our work is to study finite-sample (S)GD. Pushing our mean-field approach further is definitely a good candidate for such an analysis, even though the results will be considerably weaker (no surprise). What will be crucial in such an analysis is the relative scaling of the sample size $n$, the sequence length $L$, and the input dimension $d$. This is ongoing work.
>
> >***Sphere constraint.** How crucial is the unit-sphere constraint on $u$ and $v$? Could the analysis extend to unconstrained parameters with $L_2$ regularisation, and what obstacles would arise?*
>
> The unit-norm constraint on the student parameters $u$ and $v$ plays a number of roles.
>
> - The single-locator problem is such that what really matters is the alignment between the student parameters $(u,v)$ with the teacher parameters $(u_*,v_*)$, and not the norm. The temperature parameter $\lambda$ is of order $\sqrt d$, so that $\|u\|$ (for example), doesn't relly matter as long as it is of order $\Theta(1)$.
>
> - The main ingredients for our analysis, namely **Propositions 11** and **12** rely on $\|u\|=1$ so that the random energy levels $E_i = \lambda x_i^\top u$ are of order $\sqrt d$ as in the classical random energy model [Derrida 1982], accorss the entire optimization trajectory. Without this constraint one would have to rescale the temperature and keep tract of it. Unfornately, $L_2$-regularization can't accomplish any of this.

---

> > ### Comment · Reviewer_mLDi · 2025-08-04
> > **Reply**
> >
> > I found all of your responses clear.
> >
> > As you pointed out, scaling with respect to sample size $n$ is a challenging issue. I look forward to seeing how your analysis evolves to address more realistic scenarios in future work.

---

### Official Review · Reviewer_CnGc · 2025-07-01

**Clarity:** 2
**Significance:** 3
**Originality:** 3
**Rating:** 4
**Confidence:** 3

**Summary:**

This paper focuses on the single-locater problem (a special regression problem) to investigate the learning dynamics of self-attention with (projected) gradient flow. The most important contributions of this paper are the closed-form descriptions for the population loss and the gradient flow dynamics, which employs techniques from statistical physics and mean-field theory.

**Questions:**

1. Could the authors give how the population risk explicitly depends on $t$?

2. Could the authors explain why the stationary points would be eventually obtained by the dynamics?

3. Please refer to the questions in the Weaknesses part.

**Ethical Concerns:**

["NO or VERY MINOR ethics concerns only"]

**Final Justification:**

The authors have addressed my concerns on the convergence to stationary points.

On the other hand, as noted by me and other reviewers, the paper is hard for readers to follow due to the presentation issue and many typos as discussed in my original reivew and replies. Hence, I recommend a borderline score.

**Limitations:**

yes

**Paper Formatting Concerns:**

There is no formatting issue.

**Quality:**

2

**Strengths And Weaknesses:**

## Strengths

1. The results are generally novel and insightful to me, showing how techniques of statistical physics can be interestingly applied to solve problems regarding the learning dynamics of self-attention, though the problem being studied is a toy one. This might inspire future works to try to apply more analytical tools from physics to solve a wide range of machine learning and deep learning problems to further unveil the mystery of their success.

2. The statement of the problem setting and the corresponding motivation are clear, which might be generalized to further study the mechanisms and advantages of self-attention over other models.

## Weaknesses

1. Unfortunately, I believe that the current manuscript is hard for readers to understand, due to many typos and insufficient discussion and explanation of concepts/definitions/notations/techniques involved. I would strongly recommend that the authors improve the readability of their manuscript so that the results can be more easily appreciated, and currently I struggled to understand the details. In particular,

   - line 135-136, the authors directly present the mean-field description for the system, without any explanation on why it has to be these quantities. I would suggest the authors clearly explain the motivation for these definitions and why/how they are essential for the analysis of the system, as many readers in this venue in fact do not have a statistical physics background and it seems like that these definitions are coming from nowhere.

   - line 157 "index $\ell_{\*}\in L$ is a random datapoint", an index cannot be a datapoint; "Lemma xyz", where is this lemma?

   - There are too many different definitions, notations, and propositions spreading out the paper, and it is unclear why and how we have to have these definitions, e.g., what is the exact definition of $\simeq$ in the context of this manuscript? The readers must scroll back and forth constantly to find them when they suddenly appear in some propositions and corollaries, e.g., it is unclear why we have to define the auxiliary functions in Eq. (21) to Eq.(23).

   - The explanation for techniques is far from sufficient, e.g., line 721 how can we directly write Eq. (48)? line 755 how do we have Eq. (55)? line 787-790 what are these auxiliary functions for?

2. The population loss indeed explicitly depends on the overlaps defined in the main paper, but we still do not know how the population loss explicitly depends on $t$. As the considered problem is a toy one, I think it would be better to derive an exact solution for the population loss.

3. It is not clear why the stationary points would be eventually obtained by the dynamics (line 265-266), as we only have the formulation of the flow dynamics and there is no discussion on convergence.

---

> ### Author Rebuttal · Authors · 2025-07-31
>
> ## Response to Reviewer CnGc
> We thank the reviewer for their feedback and insightful comments.
>
> ### Main technical concerns
> >*The population loss indeed explicitly depends on the overlaps defined in the main paper, but we still do not know how the population loss explicitly depends on. As the considered problem is a toy one, I think it would be better to derive an exact solution for the population loss.*
>
> We’re not sure we understand this comment from the reviewer. The population loss $R(u,v)$ is given derived explicitly in **Proposition 1**. The formula given is valid for all unit-vectors $u$ and $v$ (the student parameters), irrespective of whether these depend on time $t$ or not. We’re happy to follow up on this if the reviewer could clarify the question.
>
> >*It is not clear why the stationary points would be eventually obtained by the dynamics (line 265-266), as we only have the formulation of the flow dynamics and there is no discussion on convergence.*
>
> Indeed, strictly speaking our statement on **Line 265 -- 266** is referring to continuous time dynamics Eqn (30) and not the discrete-time dynamics (29). The former is established in **Proposition 4**. As long as we are not worried about speed of convergence (not a concern of our analysis), the conclusions of the proposition carry over to the discrete-time dynamics (29) as long as the step-size is sufficiently small. We shall modify **Line 265 -- 266** to make this clearer.
>
> >*The readers must scroll back and forth constantly to find them when they suddenly appear in some propositions and corollaries, e.g., it is unclear why we have to define the auxiliary functions in Eq. (21) to Eq.(23).*
>
> The auxiliary functions defined in Eqn (21) -- (23) appear explicitly in some of our results (e.g., **Proposition 1**). Without those functions, the statement of those results would be rather cumbersome and unwieldy, making the manuscript even more difficult to read.
>
> >*line 787-790 what are these auxiliary functions for?*
>
> The $T_k$ functions are used to streamline the proofs our our results on classification of stationary points of the gradient-flow dynamics (**Propositions 4** and **7**). See **Appendix F**. We shall add a sentence to make this more explicit.
>
> ### Comments regarding statistical physics
> >*line 135-136, the authors directly present the mean-field description for the system, without any explanation on why it has to be these quantities. I would suggest the authors clearly explain the motivation for these definitions and why/how they are essential for the analysis of the system, as many readers in this venue in fact do not have a statistical physics background and it seems like that these definitions are coming from nowhere.*
>
> Eqn (14) -- (18) Define statistical physical constants on which the phase transitions and mean-field approximations in our theory depend. We'll add a paragraph explaining the philosophy in words to ease understanding by a broader audience. However, note that one doesn't need to know any statistical physics to follow the actual statements of the results (propositions, etc.).
>
> >*The explanation for techniques is far from sufficient, e.g., line 721 how can we directly write Eq. (48)? line 755 how do we have Eq. (55)?*
>
> Eqns (48) and (55) are the starting points for the so-called Laplace method which is a standard procedure for approximating sums by integrals and extremizing them [Lucibello and Mézard (2024)]. I'll add a section in the appendix to detail on this technique.
>
> ### Notation and typos
> - The standard notation $a(n) \simeq b(n)$ means $a(n)/b(n) \to 1$ for large $n$. For example, $n + \sqrt n \simeq n$, etc. For random quantities, the notation means $a(n)/b(n) \to 1$ in probability. We shall add a small notations paragraph in the introduction.
> - **Line 157.** "index $\ell_* \in L$ is a random data point $X=(x_1,\ldots,x_L)$" should be "index $\ell_* \in [L]$ in a random data point $X=(x_1,\ldots,x_L)$"
> - **Line 157.** "Lemma xyz" is a typo; indeed there is no Lemma xyz. Let us just proof the fact here in-line (this will be added to the manuscript): the probability of locating correctly picking the right index $\ell_*$ is $p_{\ell_*} = e^{\lambda x_{\ell_*}^\top u}/Z$, where $Z := \sum_\ell e^{\lambda x_\ell^\top u}$. From Eqn (13), we can further write
> $$
> \begin{split}
> p_{\ell_*} &= \frac{1}{1+e^{-\beta\sqrt d x_{\ell_*}^\top u}Z_{-\ell_*}} = \frac{1}{1+e^{-\left(\frac{x_{\ell_*}^\top u}{\sqrt d} - \frac{\log Z_{-\ell_*}}{\beta d}\right)\beta d}}\\
> &\simeq \frac{1}{1+e^{(cu^\top u_*-\psi)\beta d}}= \frac{1}{1+e^{(c\mu-\psi)\beta d}},
> \end{split}
> $$where we have have used Eqn (14) to write $(\beta d)^{-1}\log Z_{-\ell_*} \simeq \psi$ (this the first place where statistical  physics plays a role in our theory), and also $x_{\ell_*}^\top u/\sqrt d = cu_*^\top u + \mathcal N(0,\gamma^2/d) \simeq cu_*^\top u$ thanks to Eqn (4).

---

> ### Comment · Reviewer_CnGc · 2025-08-03
>
> Thanks for your response.
>
> 1. *"We’re not sure we understand this comment from the reviewer. The population loss $R(u,v)$ is given derived explicitly in Proposition 1. The formula given is valid for all unit-vectors $u$ and $v$ (the student parameters), irrespective of whether these depend on time $t$ or not. We’re happy to follow up on this if the reviewer could clarify the question."*
>
>     **Re:** As the model is trained by gradient flow, the dynamics depends on time $t$ (iteration count), thus both $u$ and $v$ should depend on $t$. Therefore, the population loss $R(u, v)$ also depends on $t$, and my question was **"what is this dependence?"**
>
> 2. *"Indeed, strictly speaking our statement on Line 265 -- 266 is referring to continuous time dynamics Eqn (30) and not the discrete-time dynamics (29). The former is established in Proposition 4. As long as we are not worried about speed of convergence (not a concern of our analysis), the conclusions of the proposition carry over to the discrete-time dynamics (29) as long as the step-size is sufficiently small. We shall modify Line 265 -- 266 to make this clearer."*
>
>     **Re:** I clarify that my questions was indeed for the continuous time dynamics (please see "formulation of the flow dynamics" in my original comment), not for discrete time dynamics or its speed of convergence. Here I clarify the question:  What Proposition 4 has established is the formulation of the stationary points, which does not imply that running the dynamics will eventually lead the model to converge to these stationary points. By expressing $(u, v)$ (hence $R(u, v)$) as a function of $t$, we can clearly see whether these stationary points will be attained as $t \to \infty$.
>
> 3. "Eqn (14) -- (18) Define statistical physical constants on which the phase transitions and mean-field approximations in our theory depend. We'll add a paragraph explaining the philosophy in words to ease understanding by a broader audience. However, note that one doesn't need to know any statistical physics to follow the actual statements of the results (propositions, etc.)."
>
>      **Re:** I clarify that only defining these physical constants does not sufficiently explain the motivation of the approach discussed in this paper. In addition, although one might not need to **"know any statistical physics to follow the actual statements of the results**, understanding the motivation and intuition behind the approach, e.g., why this problem can be solved with these statistical physics techniques and how phase transition can manifest in this setting, will be crucial for readers to generalize results in this paper such that the results can be appreciated by broader audience.
>
> Finally, I agree with Reviewer t3fK on that the current manuscript suffers from rushed presentation and numerous typos. Hence,  I cannot increase my score currently.

---

> ### Author Response · Authors · 2025-08-04
> **Motivation and intuition for the statistical physics (14 -- 18)**
>
> We thank the reviewer for clarifying their previous questions.
>
> The reviewer is worried about a lack of intuition surrounding the (14 -- 18). Let's provide such a intuition, by simplifying the arguments (and sacrificing some rigor/preciseless along the way). The student model can be written as $f(x;u,v) = \sum_\ell p_\ell \sigma(x_\ell^\top v) = p_{\ell_*}\sigma(x_{\ell_*}^\top v) + (1-p_{\ell_*})\sum_{\ell \ne \ell_*}q_\ell \sigma(x_\ell^\top v)$, where $p_\ell := e^{\beta\sqrt dx_\ell^\top u}/Z$ for any $\ell \in [L]$, $q_\ell := e^{\beta\sqrt d x_\ell^\top u}/Z_{-\ell_*}$ for any $\ell \in [L]\setminus\{\ell_*\}$, $Z_{-\ell_*} := \sum_{\ell \ne \ell_*}e^{\beta\sqrt d x_\ell^\top u}$, and $Z := e^{\beta\sqrt d x_{\ell_*}^\top u} + Z_{-\ell_*}$. We need to estimate the sum $f_2:=\sum_{\ell \ne \ell_*}q_\ell \sigma(x_\ell^\top v)$ over the non-secret indices $\ell \ne \ell_*$. This is a "thermal" average of the function $x \mapsto \sigma(x^\top v)$, w.r.t the Gibbs distribution $q_\ell$ induced by $L-1 \simeq L$ (which can be exponential in $d$) random energy levels $E_\ell := \sqrt d x_\ell^\top u$. Thanks to Eqn (5) of the manuscript, these energy levels are iid from $\mathcal N(0, d)$. Statistical physics tells us how to compute such averages. Unfortunately, there is no simple equivalent for such arguments in the language of classical probability and statistics. Intuitively, the average is concentrated on very few or an exponential number of energy levels, depending on the value of the inverse temperature $\beta$ (this is the so-called condensation phase-transition mentioned in **Section 4.1**). The precise formula for that average is computed in **Appendix C.1** and **C.2**, using classical techniques (Laplace's method). Unfortunately, there still isn't any **simple mechanism** for translating such arguments into the language of classical probability.
>
> Here is where statistical physics plays a role: In the limit (9), an average like $f_2$ can be analytically described via by employing the formulae (14 -- 18). This is what is done in **Proposition 11** and **12**. The calculations for the special case of linear link function $\sigma(t) \equiv t$ can be made a bit more transparent (and hopefully more intuitive/instructive to the reader) like so. Write $x_\ell^\top v= \rho x_\ell^\top u + \sqrt{1-\rho^2}z_\ell$, where $\rho:=u^\top v$ as usual, and $z_\ell \sim \mathcal N(0,1)$ is independent of $x_\ell^\top u$. Then $f_2 = \rho f_{2,1} + \sqrt{1-\rho^2}f_{2,2}$, where
> $$
> f_{2,1} := \sum_{\ell \ne \ell_*}q_\ell x_\ell^\top u,\quad f_{2,2}:=\sum_{\ell \ne \ell_*}q_\ell z_\ell.
> $$$f_{2,2}$ is noise of variance $\sum_{\ell \ne \ell_*}q_\ell^2 \le 1$. On the other hand,
> $$
> \frac{1}{\sqrt d}f_{2,1} = \frac{1}{d}\frac{\partial \log Z_{-\ell_*}}{\partial \beta} \simeq r,
> $$thanks to (15). We deduce that, $f \simeq p_* x_{\ell_*}^\top u + (1-p_*)r\rho\sqrt d$. This is the first part of **Proposition 1** for the linear link function. Using this representation for $f$ can then be used to obtain the expression for its risk given in **Corollary 1**.

---

> ### Author Response · Authors · 2025-08-04
> **Convergence to stationary point**
>
> We thank the review for clarifying their question regarding convergence. The short answer is Yes, the PGD dynamics converges to a stationary point of the risk functional $R$. More precisely, in this post, we shall prove that
> >**Claim.** *If $(u_0,v_0) \in \mathcal M$, then the PGD dynamics (29) converges to a stationary point of the risk functional $R$ (equivalently, of the PGF dynamics (30)).*
>
> The result will be included in the manuscript for completeness.
>
> ### Step 1: A Descent Lemma
> We shall need the following regularity assumption for the link function $\sigma$.
>
> >**Assumption.** The link function $\sigma$ is $C^2$ on $\mathbb R$ with $\sup_t |\sigma'(t)|$, $\sup_t |\sigma''(t) < \infty$. The case of ReLU activation needs special treatment (not provided here).
>
> One can show that on $S_{d-1}^2$, the $R$ functional
> $$
> \begin{split}
> R(u,v) &:= p^2 H_1(1,\zeta,\rho)-2pH_2(\nu,\zeta,\rho) + H_1(1,0,\rho),\\\\
> \text{with }\mu &:= u^\top u_*,\quad \nu := v^\top v_*,\quad \zeta := v^\top u_*,\quad \rho := v^\top u,\\\\
> p &:= (1+e^{(c\mu-\psi)\beta d})^{-1},
> \end{split}
> $$is $L$-smooth on $S_{d-1}^2$ for some finite $L>0$. Now, consider the following canonical the extension $r$ of $R$
> $$
> r(u,v) := R(u/\|u\|,v/\|v\|).
> $$Note that $\nabla_u r(u,v) = P_u^\perp \nabla_u R(u,v) \perp v$ and $\nabla_v r(u,v) = P_v^\perp \nabla_v R(u,v) \perp v$ for all $(u,v) \in S_{d-1}^2 \subseteq S_{d-1}^2(\delta)$. Then, one can show that for any $\delta \in (0,1]$, the functional $r$ is $L_\delta$-smooth  with $L_\delta := 4L/(1-\delta)^2$, on the tube
> $$
> S_{d-1}^2(\delta) := \{(u,v) \in \mathbb R^{2d} \mid 1 - \delta \le \|u\|,\|v\| \le 1+\delta\}.
> $$ This means that
> $$
> r(u', v') \le r(u,v) + \langle \nabla r(u,v),\Delta\rangle + \frac{L_\delta}{2}\|\Delta\|^2,\tag{2}
> $$for any $u,v,u',v'\in S_{d-1}^2(\delta)$, with $\Delta = (u'-u,v'-v) \in \mathbb R^{2d}$.
>
> Now, define $g_k = (a_k,b_k)$, where $a_k,b_k \in \mathbb R^d$ are defined by
> $$
> a_k := \nabla_u r(u_k,v_k) = P_{u_k}^\perp \nabla_u R(u,v),\quad b_k:= \nabla_v r(u_k,v_k) =P_{v_k}^\perp \nabla_v R(u_k,v_k).
> $$In (2) above, taking $u=u_k$, $v=v_k$, $u'=\tilde u_{k+1}:=u_k-sa_k$ (as in (28)), and $v':=\tilde v_{k+1}=u_k-sg_k$, we get
> $$
> \begin{split}
> r(\tilde u_{k+1},\tilde v_{k+1}) &\le r(u_k,v_k) - s\langle \nabla r(u_k,v_k),g_k\rangle + \frac{s^2 L_\delta}{2}\|g_k\|^2\\\\
> &= r(u_k,v_k) - s\|g_k\|^2 + \frac{s^2L_\delta}{2}\|g_k\|^2\\\\
> &= r(u_k,v_k) - (s-s^2L_\delta /2)\|g_k\|^2.
> \end{split}
> \tag{3}
> $$
>
> We shall now control the deviation of $(u_{k+1},v_{k+1})$ from $(\tilde u_{k+1}, \tilde v_{k+1})$. By definition, we have
> $$
> \begin{split}
> \|u_{k+1}-\tilde u_{k+1}\| &:= \|(\frac{1}{\|\tilde u_{k+1}\|}-1)\tilde u_{k+1}\| = |\|\tilde u_{k+1}\|-1|\\\\
> \|v_{k+1}-\tilde v_{k+1}\| &:= \|(\frac{1}{\|\tilde v_{k+1}\|}-1)\tilde v_{k+1}\| = |\|\tilde v_{k+1}\|-1|.
> \end{split}
> $$
> Now, because $\tilde u_{k+1} = u_k - sg_{u_k}$ with $\|u_k\|=1$ and $u_k \perp g_{u_k}$, we have
> $$
> 1 \le \|\tilde u_{k+1}\| = \sqrt{1+s^2\|g_{u_k}\|^2} \le 1+\frac{s^2}{2}\|g_{u_k}\|^2,
> $$where the last step uses the elementary inequality $\sqrt{1+a} \le 1 +a/2$ for all $a \ge 0$. Similarly, for $v_{k+1}$ we have
> $$
> 1 \le \|\tilde v_{k+1}\| =\sqrt{1+s^2\|g_{v_k}\|^2} \le 1+\frac{s^2}{2}\|g_{v_k}\|^2.
> $$On the other hand, (28) and (29) give
> $$
> u_{k+1}-\tilde u_{k+1} = (\frac{1}{\|\tilde u_{k+1}\|}-1)\tilde u_{k+1},\, v_{k+1}-\tilde v_{k+1} = (\frac{1}{\|\tilde v_{k+1}\|}-1)\tilde v_{k+1}
> $$We deduce that
> $$
> \begin{split}
> \|u_{k+1}-\tilde u_{k+1}\| &= |\|\tilde u_{k+1}\|-1| \le \frac{s^2}{2}\|a_k\|^2,\\\\
> \|v_{k+1}-\tilde v_{k+1}\| &= |\|\tilde v_{k+1}\|-1| \le \frac{s^2}{2}\|b_k\|^2.
> \end{split}
> $$Using this in (2) above gives with $u=\tilde u_{k+1}$, $v=\tilde v_{k+1}$, $u' := u_{k+1}=\tilde u_{k+1}$, $v' := u_{k+1}$, so that $\Delta = \zeta_k := (u_{k+1}-\tilde u_{k+1},v_{k+1}-\tilde v_{k+1})$ gives
> $$
> \begin{split}
> r(u_{k+1},v_{k+1}) &\le r(\tilde u_{k+1},\tilde v_{k+1}) + \langle r(\tilde u_{k+1},v_{k+1}),\zeta_k\rangle + \frac{L_\delta}{2}\|\zeta_k\|^2\\
> &=  r(\tilde u_{k+1},\tilde v_{k+1}) + \frac{s^4L_\delta}{8}\|g_k\|^2,
> \end{split}
> $$where we have used the fact that $\nabla r(\tilde u_{k+1},\tilde v_{k+1}) \perp \zeta_k$. Combining with (3) above gives
> $$
> r(u_{k+1},v_{k+1}) \le r(u_k,v_k) - (s-s^2 L_\delta/2)\|g_k\|^2 + \frac{s^4 L_\delta}{8}\|g_k\|^4.
> $$

---

> ### Author Response · Authors · 2025-08-04
> **(continued)**
>
> Now, one can show that $\|\nabla r(u,v)| \le M_\delta<\infty$ uniformly on $S_{d-1}(\delta)$. Thus, if $0<s<1/M_\delta$, then $s^4 \|g_k\|^4 = (s^2 \|g_k\|^2)s^2\|g_k\|^2 \le s^2\|g_k\|^2$, and we get
> $$
> \begin{split}
> r(u_{k+1},v_{k+1}) &\le r(u_k,v_k) - (s-s^2 L_\delta/2-s^2 L_\delta/8)\|g_k\|^2\\\\
> &=r(u_k,v_k) - (s-5s^2 L_\delta/8)\|g_k\|^2\\\\
> &\le r(u_k,v_k) - \frac{11}{16}sL_\delta \|g_k\|^2 \le r(u_k,v_k) - \frac{1}{4}sL_\delta\|g_k\|^2,
> \end{split}
> $$provided the stepsize $s$ is sufficiently small in the sense that
> $$
> 0<s<\min(1/(2L_\delta),1/M_\delta).
> $$ Noting that $r=R$ on $S_{d-1}^2 \subseteq S_{d-1}^2(\delta)$, we get the following
> >**Descent Lemma.** *If $0< s < \min(1/(2L_\delta),M_\delta)$, then*
> $$
> R_{k+1} \le R_k - \frac{sL_\delta}{4}\|g_k\|^2,\text{ with }R_k := R(u_k,v_k),
> $$where we recall that $g_k = (a_k,b_k)$, $a_k=P_{u_k}^\perp \nabla_u R(u_k,v_k)$, $b_k=P_{v_k}^\perp \nabla_v R(u_k,v_k)$.

---

> ### Author Response · Authors · 2025-08-04
> **(continued)**
>
> ### Step 2: Convergence to stationary point
> The above inequality can be rewritten as $sL_\delta \|g_k\|^2/4 \le R_k-R_{k+1}$, and summing both sides gives
> $$
> \frac{sL_\delta}{4}\sum_{k=0}^{K-1}\|g_k\|^2 \le R_0-R_K \le R_0 - R_\min < \infty,
> $$where $R_\min := \min_{u,v \in S_{d-1}}R(u,v) = 0$. We deduce that $\|g_k\| \to 0$, and so
> >$g_k \to 0$ in the limit $k \to 0$.
>
> Now,
> $$
> \begin{split}
> \|u_{k+1}-u_k\| &= \|\frac{\tilde u_{k+1}}{\|\tilde u_{k+1}\|}-u_k\| = \|\frac{u_k-sa_k}{\|\tilde u_{k+1}\|}-u_k\|\\\\
> &= \|\frac{1-\|\tilde u_{k+1}}{\|\tilde u_{k+1}\|}u_k-\frac{s}{\|\tilde u_{k+1}\|}a_k\|\\\\
> &\le (|1-\|\tilde u_{k+1}\|| + s\|a_k\|)\frac{1}{\|\tilde u_{k+1}\|}\\\\
> &\le \frac{s^2}{2}\|a_k\|^2+s\|a_k\|\text{ because }\|\tilde u_{k+1}\| \ge 1\\\\
> &\le \frac{s}{2}\|a_k\| + s\|a_k\|\text{ because } s\|a_k\| \le 1\\\\
> &= \frac{3}{2}s\|a_k\|.
> \end{split}
> $$Anologously, we get $\|v_{k+1}-v_k\| \le (3/2)s\|b_k\|$. Combining gives
> $$
> \|u_{k+1}-u_k\|^2 + \|v_{k+1}-v_k\|^2 \le \frac{9}{4}s^2\|g_k\|^2 \to 0.
> $$
> >We deduce that the PGD iterates $(u_k,v_k)$ form a Cauchy sequence in $S_{d-1}^2$. Due to completeness of $S_{d-1}^2$, this sequence has a limit $(u_\infty,v_\infty) \in S_{d-1}^2$.
>
> We now show that $(u_\infty,v_\infty)$ is a stationary point of the risk functional $R$.
>
> >For simplicity of presentation, we focus on the case of linear link function $\sigma(t) \equiv t$.
>
> Thanks to **Proposition 3**, if $(u_0,v_0) \in \mathcal M$, then $(u_k,v_k) \in \mathcal M$ for all $k$ and one computes,
> $$
> \begin{split}
> a_k&=(I-u_ku_k^\top)\nabla_u R(u_k,v_k)=(I-u_ku_k^\top)T_1u_*=(u_*-\mu_k u_k)T_1,\\\\
> b_k&=(I-v_kv_k^\top)\nabla_v R(u_k,v_k)=(I-v_kv_k^\top)T_2 v_* = (v_*-\nu_k v_k)T_2,
> \end{split}
> $$where $T_j=T_j(\mu_k,\nu_k,0)$ for $j=1,2,3$, as defined in **Appendix E.1**, with $\mu_k := u_k^\top u_*$ and $\nu_k := v_k^\top v_*$. Note that we have used the fact that $T_3(\mu_k,\nu_k,0) = 0$ thanks to the equation just because **Line 882** in the proof of **Proposition 3**. We deduce that
> $$
> \|g_k\|^2 = \|a_k\|^2 + \|b_k\|^2 = (1-\mu_k^2)T_1^2 + (1-\nu_k^2)T_2^2.
> $$
> Thus, the limit point $(u_\infty,v_\infty)$ is such that
> $$
> (1-\mu_\infty^2)T_1(\mu_\infty,\nu_\infty,0)=0,\quad (1-\nu_\infty^2)T(\mu_\infty,\nu_\infty,0)=0.
> $$This is precisely the characterization of stationary points risk functional $R$ (equivalently, of the PGF dynamics (30)) established in **Proposition 7**. This concludes the proof of the claim. $\quad\quad\Box$

---

> > ### Author Response · Authors · 2025-08-04
> >
> > We hope the above detailed responses have addressed the remaining concerns of the reviewer. We'd be greatful if they would kindly increase their score accordingly. Thanks in advance.
> >
> > We are happy to answer to any further questions / concerns.

---

> > > ### Comment · Reviewer_CnGc · 2025-08-06
> > >
> > > Thanks for your detailed reply.
> > >
> > > ## Regarding the proof for convergence to stationary points
> > >
> > > 1. I believe that the proof for the convergence to stationary points, which should be an important part for the analysis of the optimization dynamics, should have been posted in the original rebuttal, not the follow-up comment, as I believe that my original comment was clear on this aspect and the reason why I clarified my question in my reply was because the authors did not reply properly at the first place. The proof posted in this late stage cannot allow me to review it sufficiently.
> > >
> > > 2. The proof strategy seems correct to me, but I cannot verify the correctness in detail at this late stage. Anyway, my concern on this aspect could be addressed by this proof, and I believe the corresponding discussion should be included in the revision.
> > >
> > > 3. I'm a bit confused about why the authors decided to prove the case for the discrete dynamics when the original discussion in the manuscript was for the continuous dynamics. Based on my understanding, they are not equivalent as implied by the authors in their reply. The flow of the paper should be consistent: both the analysis for the convergence to stationary points and that for the dynamics should be conducted either for the discrete time or continuous time (or both) at the same time.
> > >
> > >
> > > ## Dependence on $t$
> > >
> > > I'm not clear about why the authors still did not choose to answer the question of the dependence of the population loss on the time $t$, which could show the convergence to the stationary points at the same time.
> > >
> > > ## Presentation
> > >
> > > As noted by me and other reivewers, the current manuscript at least needs a major revision to improve the readability and presentation.
> > >
> > > In summary, the authors addressed my technical concerns, but the presentation needs to be improved significantly. I will increase my score for the efforts of the authors in the discussion period.

---

> > > > ### Author Response · Authors · 2025-08-06
> > > >
> > > > >*I believe that the proof for the convergence to stationary points, which should be an important part for the analysis of the optimization dynamics, should have been posted in the original rebuttal, not the follow-up comment, as I believe that my original comment was clear on this aspect and the reason why I clarified my question in my reply was because the authors did not reply properly at the first place. The proof posted in this late stage cannot allow me to review it sufficiently.*
> > > >
> > > > We are happy that our long comment answers the reviewers question on convergence, modulo the details. We are sorry that we main point of the reviewer's initial question in our rebuttal. This will be integrated into the manuscript, alongside a discussion.
> > > >
> > > > >*I'm a bit confused about why the authors decided to prove the case for the discrete dynamics when the original discussion in the manuscript was for the continuous dynamics.*
> > > >
> > > > Because it turns out the convergence of PGD (which is the thing that actually gets implemented on a computer) as defined in Eqn (28) and (29) can be done directly (as shown in previous post), with a quantitative upper-bound on the required step size $s$. This is a much stronger result. We shall include this discussion in the manuscript.
> > > >
> > > > >*I'm not clear about why the authors still did not choose to answer the question of the dependence of the population loss on the time, which could show the convergence to the stationary points at the same time.*
> > > >
> > > > Because there is no explicit formula of R as a function of R, like $R_t = 1/\log t$, etc. In general there is never such an explicit formula. Quantities of interest (here, the population risk $R$) are given only implicitly in terms of the equations defining the dynamics, here the PGD Eqn (29) (resp. (30) for PGF), like so $R_k = R(u_k, v_k)$ where $u_k$ and $v_k$ evolve according to (29) (resp. (30)). Fortunately, we don't need an explicit formula for $R$ in terms of $k$ (or $t$) to prove convergence of the dynamics. We hope this answers the reviewer's question about explicit dependence of $R$ on $t$ (or $k$). Sorry if this is not the case; happy to follow up.
> > > >
> > > > >*As noted by me and other reviewers, the current manuscript at least needs a major revision to improve the readability and presentation.*
> > > >
> > > > >*In summary, the authors addressed my technical concerns, but the presentation needs to be improved significantly. I will increase my score for the efforts of the authors in the discussion period.*
> > > >
> > > > We shall polish all the rough edges, fix the typos, and integrate the reviewers comments in the final version. We thank the reviewer once again for an insightful discussion and for promising to raise their rating (the updated rating is not not yet visible to us) of our work.

---

### Official Review · Reviewer_t3fK · 2025-07-02

**Clarity:** 3
**Significance:** 2
**Originality:** 2
**Rating:** 3
**Confidence:** 3

**Summary:**

This paper closely follows the work of Marion et al. (2025) and focuses on the single-location regression problem. The authors provide a comprehensive and formal definition of the problem and extend the analysis to include softmax-based attention mechanisms. They derive an analytic expression for the risk and use gradient-based analysis to demonstrate that the optimal relation can be achieved through learning.

**Questions:**

The theoretical derivations presented in the paper are impressive and technically well-executed. However, many of the results feel like intermediate steps that ultimately lead nowhere concrete. It is unclear how these derivations contribute to a deeper understanding of model behavior or guide practical model design. Strengthening the connection between the theoretical results and their implications would greatly enhance the impact of the work.

**Ethical Concerns:**

["NO or VERY MINOR ethics concerns only"]

**Final Justification:**

The authors clarified their motivation and contributions in the rebuttal. However, the work appears to be a follow-up with mostly incremental theoretical extensions over prior results. The presentation remains difficult to follow, and the lack of empirical validation limits the perceived impact. While I recognize that others may see theoretical value here, I remain unconvinced and lean toward a borderline reject.

**Limitations:**

yes

**Quality:**

3

**Strengths And Weaknesses:**

Strengths:
- The paper provides a comprehensive and rigorous theoretical walkthrough.

Weaknesses:
- The motivation for this work is unclear. In Marion et al. (2025), the analysis is grounded in a meaningful setup where one word decisively determines the output, making the theoretical insights directly relevant to real-world model behavior. In contrast, the current paper lacks a clear justification for why the theoretical results are useful or what practical implications they have. As a result, it is difficult to understand what is gained from this analysis or how it informs model design or learning dynamics in more realistic settings.
- Too many typos. Like here on top in the title, an unintended backslash (\) appears, and in the paper on line 241 the phrase “shall shall” is repeated.

---

> ### Author Rebuttal · Authors · 2025-07-31
>
> ## Response to Reviewer t3fK
> We thank the reviewer for their feedback. We shall correct the typos mentioned by the reviewer. Below, we provide a detailed bullet-point response to the main comments of the reviewer.
>
> >*The motivation for this work is unclear. In Marion et al. (2025), the analysis is grounded in a meaningful setup where one word decisively determines the output, making the theoretical insights directly relevant to real-world model behavior. In contrast, the current paper lacks a clear justification for why the theoretical results are useful or what practical implications they have.*
> >
> >*As a result, it is difficult to understand what is gained from this analysis or how it informs model design or learning dynamics in more realistic settings.*
>
>
> As in [Marion et al. 2024], the mission of our work is to contribute to the theoretical understanding of learning in transformers and optimization dynamics. Our work considers actual **softmax** self-attention layers (unlike the reference work which considers pointwise attention), and considerably extends the range and scope of the findings (global convergence to teacher parameters, etc.).
>
> The setup of our work and our main contributions are well spelt out clearly in the paper see **Introduction** and **Concluding Remarks** sections. Let us repeat a summary thereof here:
>
>
> **[Marion et al. 2024]** considers solving the so-called single-locator regression problem (described in the introduction of our paper; see **Figure 1**) via *student model* which is simplified self-attention layer where the softmax is replaced by a **pointwise activation** function $\theta$, namely $f(x;u,v) := \sum_{\ell=1}^L p_\ell x_\ell^\top v$ for any input $x \in \mathbb R^d$, with $p_\ell := \theta(\lambda x_\ell^\top u)$, as in Eqn (1). Here $L$ is the sequence length and $d$ is the input dimension. They show that this simplified model can be trained to recover the oracle / teacher parameters $(u_*,v_*)$. They also study the optimization dynamics and show that convergence to $(u_*,v_*)$ is ensured assuming gradient descent (GD) is initialized on a certain manifold $\mathcal M$ formed by parameters in a peculiar configuration. This is local convergence.
>
> **Our work** extends [Marion et al; 2024] in many directions:
>
> - **1. Softmax attention and general link function.** Keeping student model same as in [Marion et al. 2024], we change the student softmax attention (which is what is actually used in real-world transformers!): $f(x;u,v) := \sum_{l=1}^L p_\ell \sigma(x_\ell^\top v)$, where $(p_1,\ldots, p_L)$ is now the softmax of $(x_1^\top u,\ldots,x_L^\top u)$ at a temperature $\lambda \asymp \sqrt d$, and $\sigma$ is a generic *link function* which can be linear or non-linear (e.g quadratic, ReLU, etc.). Note that these are two separate extensions of [Marion et al. 2024]: the softmax attention and (2) the general link function $\sigma$. Importantly, softmax attention maps more faithfully to what real world transformers (as noted by the other reviewers, e.g. **Reviewer mLDi**).
>
> - **2. Realizability and local convergence.** For this considerably more complicated (but realistic) student model, we show  (**Proposition 2**) that optimal parameters are in fact the teacher parameters $(u_*,v_*)$, and that GD on square loss finds it (**Propositions 5, 6, 8, 9**). For general non-linear link function $\sigma$, our convergence is local, as in [Marion al. 2024]
>
> - **3. Global convergence.** In the special case of linear link function $\sigma(t) \equiv t$ (considered in [Marion et al. 2024]), we show in **Proposition 4** that for our softmax-attention student model, GD converges to $(u_*,v_*)$ irrespective of the initialization.
>
> - **4. Very long sequences.** The results of [Marion et al. 2024] were established on the very restrictive condition of short sequence lengths: $L=o(1)$, i.e $L/d \to 0$. In contrast, our results for a vast range of sequence lengths $L$, upt o exponential in $d$, i.e $\log L=O(d)$. Such long sequences capture the situation in real world NLP / LLM workloads (a desirable property of our theory, also noted by **Reviewer mLDi**), and marks an exponential increase compared to [Marion et al. 2024]. See **Remark 4** of our manuscript.
>
> ### Reconsideration of the reviewer's score
> >We hope the above points provide the reviewer with a better understanding and appreciation of our contributions, and they increase the score accordingly. Currently, the reviewer scores our paper with a $1$ (**Strong Reject**), which is the **lowest possible score**, only reserved for papers with "well-known results or unaddressed ethical considerations" and **dismissive** and **very unfair**, which is not the case of our paper which makes a broad range of contributions to a fundamental problem. Thank you in advance for your understanding and consideration. We are happy to answer any questions the reviewer might have regarding our work.
>
> >Regarding the numerous typos which have obstructed the reviewer's understanding and appreciation of our work, it has been very difficult to keep track of everything as a **sole author**. I sincerely apologize. These typos will be carefully corrected. Thanks in advance for your understanding and consideration.

---

> > ### Comment · Reviewer_t3fK · 2025-08-02
> >
> > Thank you for the clarification. My initial score was mainly due to the rushed presentation and numerous typos, which made the paper feel like a low-quality draft. While the rebuttal helped clarify the intended contributions, I still find the practical motivation and relevance limited. I’ve slightly adjusted my score to reflect the improved clarity, but my main concerns remain.

---

> ### Author Response · Authors · 2025-08-04
>
> >*Thank you for the clarification. My initial score was mainly due to the rushed presentation and numerous typos, which made the paper feel like a low-quality draft. While the rebuttal helped clarify the intended contributions, I still find the practical motivation and relevance limited. I’ve slightly adjusted my score to reflect the improved clarity, but my main concerns remain.*
>
> We'd be happy to respond to the reviewer if they can be more precise about what concerns of theirs have not been answered by our previous response. We find the statement "my concerns remain" to be uninformative and unfair to the authors. Our previous response has attended to the reviewers concerns in a detailed bullet-point format. The concerns about the motivation of our work have been clairly responsed to. If the reviewer has any follow up questions, we would be happy to follow up on those.
>
> >Once again, a rating of **1** (**Strong Reject**), is the lowest possible score, only reserved for papers with "well-known results or unaddressed ethical considerations"
>
> We believe that our work contains none of the above. Our work makes a non-trivial contribution to an important problem: theoretical foundations  of learning and optimization in transformers.
>
> Thanks

---

### Official Review · Reviewer_Mp31 · 2025-07-02

**Clarity:** 4
**Significance:** 3
**Originality:** 3
**Rating:** 4
**Confidence:** 1

**Summary:**

This paper explores a theoretical understanding of softmax self-attention, a key component of transformer models. It investigates how softmax self-attention functions in the context of the single-location regression problem, where the goal is to locate a specific token in a sequence. By using concepts from statistical physics, the authors provide exact analytic expressions for the population risk in softmax self-attention models, and they analyze the optimization dynamics under gradient descent. They also prove that under broad conditions, the dynamics converge to the optimal solution.

**Questions:**

- The authors mention that this work removes critical initialization assumptions that previous works (e.g., Marion et al.) relied on. Could you elaborate on how this will impact the practical training of transformer models, especially when working with models that involve multiple layers or more complex structures?

- While the toy problem offers theoretical clarity, how do you envision the insights from this work being extended to more complex transformer models or real-world datasets? Are there any assumptions that would need to be modified for those contexts?


- Can the optimization dynamics described here be generalized to other attention mechanisms (e.g., linear attention or variations in the attention head)?

**Ethical Concerns:**

["NO or VERY MINOR ethics concerns only"]

**Final Justification:**

All of the concerns are addressed, and I will keep my score.

**Quality:**

4

**Strengths And Weaknesses:**

#### Strengths

- The paper offers a clear and precise mathematical framework, drawing from statistical physics (Random Energy Model) and mean-field theory. The rigorous treatment of softmax self-attention is a valuable contribution to understanding this crucial model component.

- Impactful for Transformer Models: By improving our understanding of self-attention, this work has the potential to influence the design and optimization of transformers, which are foundational to modern AI systems like LLMs.

- The use of well-structured propositions and corollaries, with complete proofs in the appendices, makes the paper both rigorous and accessible to those familiar with the theory.

#### Weaknesses

- The work is quite theoretical, with few experimental results provided. While this is understandable given the focus of the paper, empirical validation would have added strength to the claims.

- The focus is on a toy problem (single-location regression), which is useful for theoretical insights but may not directly translate to larger, more complex tasks that real-world transformers handle.

---

> ### Author Rebuttal · Authors · 2025-07-31
>
> ## Response to Reviewer Mp31
>
> ### Impact and Extensions
> **Potential impact on practical training of transformers.**
> The optimization dynamics of transformer-based models is still not well-understood, and for good reason: an attention layer is much more complex than an CNN or dense / full-connected layer, say. This is mostly due to the softmax non-linearity, which renders the system highly non-separable. The same thing which makes transformers highly performant (compared to traditional layers like dense layers and CNN layers), makes their analysis nightmarish.
>
> Our work contributes to a very limited (but growing) pool of theoretical works which attempt to explain the mysteries of optimization of and learning in transformer models. Our results classification of stationary points, and on the robustness of the initialization provides non-trivial much desired insights into the robustness learning into our simple transformer model. transformer models.
>
>
> **Extension to more complex transformer.** Yes, we strongly believe that the key technical ideas in our work can be leveraged to attack the case of more complex transformer architectures (multiple layers, multiple heads, etc.). In particular, with some additional work it should be possible to develop versions of **Propositions 11** and **12** (the main technical ingredients in our work) which are adapted to such settings. This is left for feature work alongside the other things mentioned under the **Concluding Remarks** section of the manuscript.
>
> **Linear attention.** Note that, the case of linear attention is drastically easier to handle, and has already been done in the literature (e.g. see [Yue et al. 2024], [Yedi et al. 2025])
>
> ### Additional References
> - [Yedi et a. (2025)] "Training Dynamics of In-Context Learning in Linear Attention"

---

> > ### Comment · Reviewer_Mp31 · 2025-08-05
> >
> > Thanks for your detailed reply, I have no other questions for the time being. From the other reviews, it seems the readability can be improved in the revised version. I am not that familiar with this field so I did not check the mathematical descriptions carefully. But I think an excellent paper should be easy to read for most researchers. Some simple descriptions for complex mathematical concepts may be required, if necessary. I hope the authors to revise these typos in the revised version.

---

### Decision · Program_Chairs · 2025-09-17

**Decision:**

Accept (poster)

**Comment:**

This paper presents a rigorous theoretical analysis of a single-layer softmax self-attention network. By leveraging tools from statistical physics, specifically the random energy model, the authors derive an exact mean-field theory for the "single-location regression task." This allows them to obtain a closed-form expression for the population risk and precisely characterize the learning dynamics of gradient descent. A key contribution is proving the global convergence to the optimal solution for linear link functions without requiring special initialization unlike prior work. The analysis also extends to long-sequence regimes, making it more relevant to modern NLP applications.

The application of statistical physics methods to derive an exact mean-field theory for a softmax attention layer is a novel and significant contribution that goes beyond the simplified pointwise activations studied in previous work. The mathematical treatment is rigorous and provides some insights into the model's behavior. Moreover, the paper establishes a clear phase transition for learnability and a global convergence guarantee for the linear case, which brings a valuable insight to the community.

Although the reviewers acknowledged these contributions, they raised concerns on its presentation quality including specialized notation, several typos, and insufficient explanations to enhance intuition for non-expert readers. I think the authors should address these issued before it will be published and the paper would benefit from a careful revision to improve its accessibility. However, its core technical contributions are novel and significant in the community. Therefore, I recommend acceptance.